# Spatially resolved hourly traffic emission over megacity Delhi using advanced traffic flow data

Akash Biswal[1, 2], Vikas Singh[1*], Leeza Malik[3], Geetam Tiwari[4], Khaiwal Ravindra[5], Suman Mor[2]

[1]National Atmospheric Research Laboratory, Gadanki, AP, 517112, India

[2]Department of Environment Studies, Panjab University, Chandigarh, 160014, India

[3]Department of Civil Engineering, Indian Institute of Technology (Indian School of Mines), Dhanbad, Jharkhand 826004, India

[4]Transportation Research and Injury Prevention Programme, Indian Institute of Technology Delhi, Hauz Khas, New Delhi 110016, India

[5]Department of Community Medicine and School of Public Health, Post Graduate Institute of Medical Education and Research (PGIMER), Chandigarh 160012, India

*Correspondence*: Vikas Singh (vikas@narl.gov.in)

**Abstract.** This paper presents a bottom-up methodology to estimate multi-pollutant hourly gridded on-road traffic emission using advanced traffic flow and speed data for Delhi. We have used the globally adopted COPERT (Computer Programme to Calculate Emissions from Road Transport) emission functions to calculate the emission as a function of speed for 127 vehicle categories. At first the traffic volume and congestion (travel time delay) relation is applied to model the 24-hour traffic speed and flow for all the major road links of Delhi. The modelled traffic flow and speed shows an anti-correlation behaviour having peak traffic and emissions in morning-evening rush hours. We estimated an annual emission of 1.82 Gg for PM (Particulate Matter), 0.94 Gg for BC (Black Carbon), 0.75 Gg for OM (Organic Matter), 221 Gg for CO (Carbon monoxide), 56 Gg for $NO_x$ (Oxide of Nitrogen), 64 Gg for VOC (Volatile Organic Carbon), 0.28 Gg for $NH_3$ (Ammonia), 0.26 Gg for $N_2O$ (Nitrous Oxide) and 11.38 Gg for $CH_4$ (Methane) for 2018 with an uncertainty of 60%- 68%. The hourly emission variation shows bimodal peaks corresponding to morning and evening rush hours and congestion. The minimum emission rates are estimated in the early morning hours whereas the maximum emissions occurred during the evening hours. Inner Delhi is found to have higher emission flux because of higher road density and relatively lower average speed. Petrol vehicles dominate emission share (> 50%) across all pollutants except PM, BC and $NO_x$, and within them the 2W (Two-wheeler motorcycles) are the major contributors. Diesel fuelled vehicles contribute most of the PM emission. Diesel and CNG vehicles have a substantial contribution in $NO_x$ emission. This study provides very detailed spatio-temporal emission maps

for megacity Delhi, which can be used in air quality models for developing suitable strategies
to reduce the traffic related pollution. Moreover, the developed methodology is a step forward
in developing real-time emission with the growing availability of real-time traffic data. The
complete dataset is publicly available on Zenodo at https://doi.org/10.5281/zenodo.6553770
(Singh et al., 2022).
**Key words:** COPERT, Multi-pollutant emission inventory, Diurnal Emission, Road transport,
Exhaust emissions, Air quality.

## 1 Introduction

Exposure to vehicular emissions poses a greater risk to the air quality and human health (Lipfert
et al., 2008; Salo et al., 2021, GBD 2021). On-road transport is the major contributor to the
ambient air pollution and greenhouse gas emissions in urban areas, mainly near roads (Singh
et al., 2014), therefore they are an important component of the local air quality management
plans and policies (Gulia et al., 2015; DEFRA, 2016; NCAP, 2019; Sun et al., 2022). The actual
traffic emission depends on several dynamic factors, such as emission factors, traffic volume,
speed, vehicle age, road network and infrastructure, road type, fuel, driving behaviour,
congestion etc. (Pinto et al, 2020; Jiang et al., 2021; Deng et al., 2020). Traffic emission
modelling has evolved and improved over recent years, however gaps still exist because of the
complexity and data involved in the emission inventory development. Moreover, the reliability
of the emission decreases further when the emissions are spatially and temporally segregated
(Super et al., 2020, Osses et al., 2021). There are differences in the reliability of emission
inventories of developed and developing countries because of lack of space-time input data in
developing countries (Pinto et al, 2020). The uncertainty associated with emission inventory is
further propagated in air quality models making mitigation studies more challenging, mainly
for developing countries such as India which is already facing air pollution issues (Pandey et
al., 2021).
India is among the top 10 economies (6th GDP rank) in the world in 2020 (GDP, 2020) and is
recognized as a developing country. The population and economic growth have led to dense
urbanisation with poor air quality in cities (Ravindra et al., 2019; Liang et al., 2020; Singh et
al., 2021). India hosts 22 cities among the top 30 polluted cities in the world (IQAIR, 2020).
The national capital of India, Delhi, has pollution levels exceeding NAAQS and WHO
guideline values (Singh et al., 2021). Earlier studies have estimated on-road traffic as the major
local contributor to Delhi pollution (CPCB 2010; Sharma et al., 2016) along with long range
transport sources associated with stubble burning and dust leading to severe pollution episodes
(Liu et al., 2018; Bikkina et al., 2019; Khaiwal et al., 2019; Beig et al., 2020; Singh et al.,

70  2020).

Delhi traffic exhaust (tailpipe) emissions have been studied extensively using different
methodology for years. The emissions estimated by various studies show large variations (see
comparison tables in Guttikunda and Calori, 2013; Goyal et al., 2013; Sharma et al., 2016;
Singh et al., 2018, and in Table 6) suggesting that the emissions have large uncertainties
associated with the method and data used.   Most of the studies adopted a bottom-up
methodology to calculate the total emission over Delhi based on the registered vehicles and
average vehicle kilometre travelled (VKT) multiplying with emission factors. A few studies
(eg., Sharma et al., 2016; Singh et al., 2018, 2020) use an on-road traffic flow approach where
emission is estimated for each line source (road link) then spatially segregated (Tsagatakis et
al., 2020, Spatial of emissions methodology). CPCB (2010), Goyal et al. (2013) further
spatially desegregated the total emissions to 2 km × 2 km resolution but the method of gridding
is not discussed in detail. Sharma et al. (2016) and TERI (2018) also estimated 2km × 2km and
4 km × 4 km gridded emission respectively, by adopting a per grid traffic flow method.
Guttikunda and Calori (2013) estimated the 1 km × 1 km gridded emission by disaggregating
the net emission using various spatial proxies like gridded road density.  Though these studies
with coarser resolution are helpful for identifying the emission hotspots but they lack actual
traffic flow information disaggregated by road type and vehicle type within the grids.
Moreover, their emission estimate shows large variations. For e.g., Das and Parikh (2004) and
Nagpure et al. (2013) estimated traffic emission using VKT methodology for the same base
year 2004, however their estimates varied by a factor of two or more. The annual emission
estimate around year 2010 by CPCB (2010), Sahu et al. (2011, 2015), Goyal et al. (2013),
Guttikunda and Calori (2013) and Singh et al. (2018) varied considerably from 3.5 Gg to
~15Gg for PM emission and 30 Gg to 200 Gg for $NO_x$ emissions. The VKT based estimation
approaches (Nagpure et al., 2013; Goel et al., 2015a; TERI 2018) tend to estimate higher
emission compared to the traffic flow methodology (Sharma et al., 2016; Singh et al., 2018).
A 40% increase in $PM_{2.5}$ emission in 2018 as compared to 2010, is reported by SAFAR (2018)
attributed to the increase in vehicular growth.

Most of the studies for Delhi use EFs developed by ARAI (Automotive research association of
India, ARAI; 2008) and a few studies have used EFs from IVE (International Vehicular
Emission Model by USEPA, Davis et al., 2005) and COPERT (Ntziachristos et al., 2019).
ARAI EFs are measured in laboratory conditions, operating the vehicles in variable speed
known as the Indian driving cycle (IDC, ARAI., 2008). The IVE emission factors are a function
of the power bins of the vehicle engine, whereas in COPERT emission factors are a function
of average vehicle speed, vehicle technologies, estimated pollutants, correction methods, and
adjustments to local conditions. (Cifuentes 2021).  Goyal et al. (2013) used the IVE model to
estimate the traffic emission over Delhi for the year 2008 and also studied the diurnal emission
at a specific location.  However, the study is limited to a fixed major traffic intersection only.
Kumari et al. (2013) used the COPERT-3 emission factor to estimate emission for Indian cities,
focusing on the multi-year (19991-2006) evolution of vehicular emission. However, this study
estimates the total emissions based on registered vehicles and does not provide spatial
segregation. COPERT Tier-3 emissions have been used for comparison with real-world
measured emission factors (Jaikumar et al., 2017; Choudhary and Gokhale, 2019). Jaikumar et
al. (2017) identified vehicle idling is the major factor in the deviation between model-based
estimation and measured emission as the vehicles spend 20% of their time in idling mode.
The traffic volume and speed information over each road are vital for accurate emission
estimation. The data over Delhi has been very limited, therefore studies have used the VKT
approach which uses the number of registered vehicles to estimate the emission. To the best of
our knowledge, despite several studies for Delhi, none of the studies have studied Delhi
emissions using advanced and detailed traffic data and speed based EFs to estimate the hourly
gridded emissions at high resolution. Moreover, most of the studies are limited to the estimation
of PM, $NO_x$, CO and HC only. The availability of recent detailed traffic data and speed volume
relation (Malik et al., 2018; 2021) as a part of the Transportation research and injury prevention
programme (TRIPP) of IIT Delhi provides an opportunity to estimate and improve the
emissions over Delhi. To the best of our knowledge, this is the first study of its kind which
considers advanced traffic flow data and estimates the hourly multi-pollutant emissions as a
function of speed.
In this study, we have adopted a globally accepted methodology based on COPERT-5 Tier3 to
estimate the hourly gridded emission for Delhi at high resolution for 2018. COPERT EFs have
been used in many studies Alamos et al. (2021) for Chile, Mangones et al. (2019) for Bogota
Cifuentes et al. (2021) for Manizalesto, Wang et al. (2010) for Chinese cities, Vanhulsel et al.
(2014) for Belgium, Tsagatakis et al., (2019) for the national emission inventory over the UK
and also has been used by many around the globe (https://www.emisia.com/utilities/copert/).
We combine advanced traffic volume and speed data (TRIPP, Malik et al., 2018) with speed
based emission factors to calculate the emissions. The methodology considers different vehicle
types, fuel type, engine capacity, emission standard and other key parameters such as
congestion to estimate the emission for each road. We estimate the emission of particulate and
gaseous pollutants namely PM (Particulate Matter), BC (Black Carbon), OM (Organic Matter),
CO (Carbon Monoxide), $NO_x$ (Oxides of Nitrogen), VOC (Volatile Organic Compound), $NH_3$
(Ammonia) and greenhouse gases, $N_2O$ (Nitrous Oxide) and $CH_4$ (Methane). Most of the PM
(~98%) from the vehicular exhaust is $PM_{2.5}$ (ARAI 2008; Pant and Harrison 2013). We study
the diurnal and spatial variability in the emission and identify the most polluting vehicle
category, hotspots and the time when traffic emissions are highest. This study provides very
detailed spatio-temporal emission maps for megacity Delhi that can be used in air quality
models for developing suitable strategies to reduce the traffic related pollution. Moreover, the
developed methodology is also a step forward in developing real-time emission models in the
future with growing availability of real-time traffic data.

**2 Methodology:**
We estimated the emissions for 2018 over the National Capital Territory (NCT) of Delhi having
an area of 1483 $km^2$ (Fig. 1) and a population of 16.8 million (Census, 2011). The domain has
been further divided into three regions (viz. Inner, Outer and Eastside), as shown in Fig. 1, to
study the spatial variation in the emissions. Inner Delhi constitutes the major business hubs and
workplaces within the ring road and the Outer is the area away from the ring road whereas
the Eastside is the east part beyond the Yamuna River.
A bottom-up emission methodology has been adopted and a python-based model has been
developed to estimate gridded hourly emissions of major pollutants over an urban area. The
model estimates emission of PM, BC, OM, CO, $NO_x$, VOC, $NH_3$, $N_2O$ and $CH_4$. The model
uses hourly traffic activity and COPERT based emission factors as a function of hourly speed
for each road link across Delhi. The major vehicle categories include 2W (Two wheeler motor
bikes), 3W (Auto rickshaws), CAR (Passenger cars), BUS (Buses), LCV (Light Commercial
Vehicles) and HCV (Heavy Commercial Vehicles).

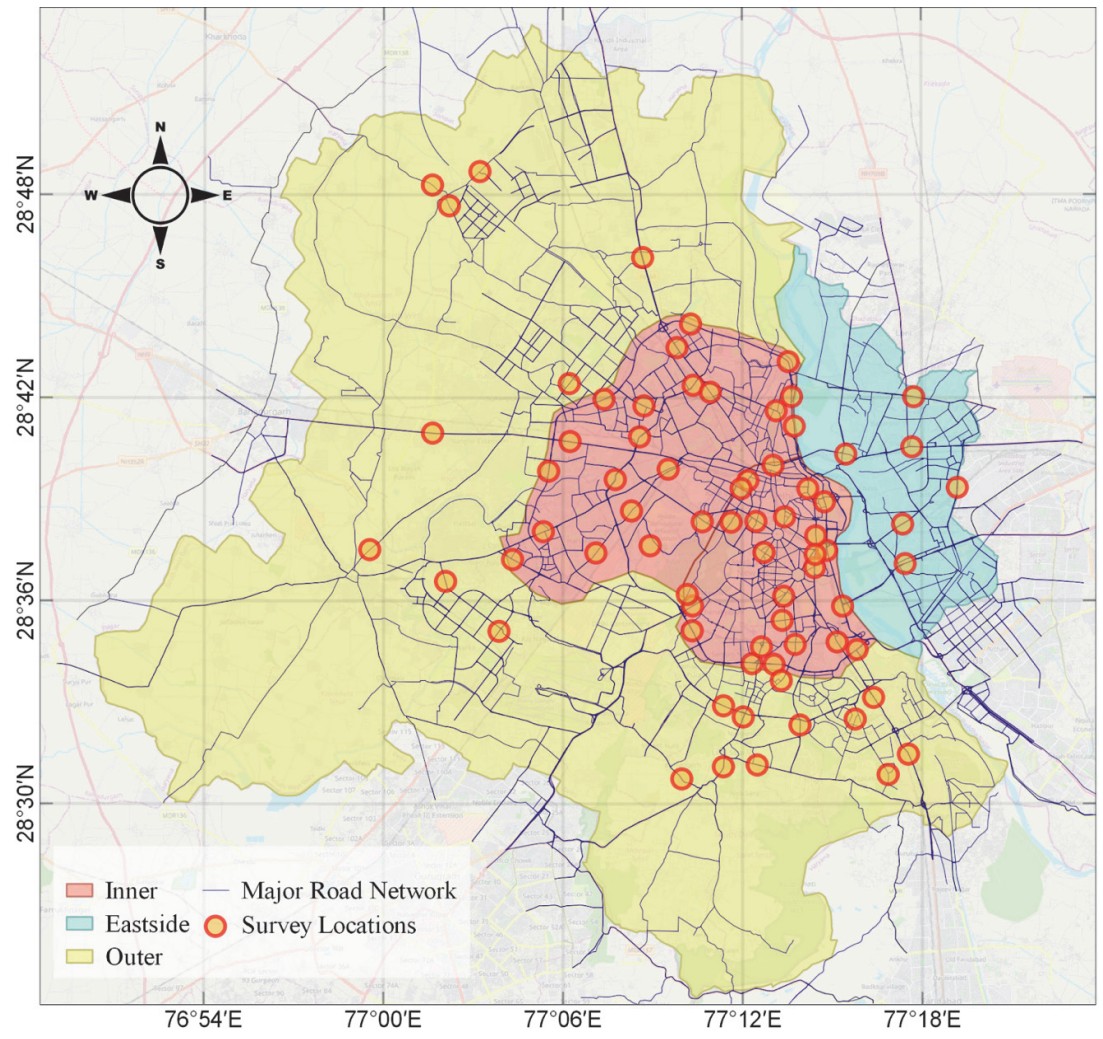


Figure 1. Map showing the study domain with TRIPP survey locations and the major road links
over Delhi. The domain is segregated to three regions (Inner, Eastside and Outer) shown in
different colours. The background map is from https://www.openstreetmap.org/; ©
OpenStreetMap contributors 2022. Distributed under the Open Data Commons Open Database
License (ODbL) v1.0.

**2.1 Traffic Activity**
Classified traffic volume and speed study of Delhi (Malik et al., 2018) provides traffic count
and speed for the roads of Delhi based on the Traffic volume and speed measurements
conducted at 72 locations (Fig. 1) over Delhi in the year 2018 as a part of Transportation
research and injury prevention programme (TRIPP) of IIT Delhi. We will refer to this dataset
as TRIPP data from now on. TRIPP provides hourly traffic from 08:00-14:00 hours for eight
fleet types (2W, 3W, Cars, Buses, Minibuses, HCV, LCV and NMV: Non-motorized vehicle)
on over twelve thousand major road links over Delhi (Malik et al., 2018). These road links are
further classified into five road classes (RClass1 to RClass5) based on the width of the road
(Table S2). More detail of TRIPP traffic flow and its methodology is available elsewhere
(Malik et al., 2018; Malik et al., 2021). As the TRIPP data is only available for 0800-1400
hours, we use speed-flow-density relationship by Malik et al. (2021) to estimate the hourly
traffic for each road link in Delhi.
**2.1.1 Generating traffic flow from congestion**
The relation between traffic volume and congested speed has been studied extensively using
Greenshield model, the Greenberg model and the Underwood model (Wang et al., 2014;
Hooper et al., 2014) and used by many studies (Jing et al., 2016; Yang et al., 2019) to estimate
the traffic from the congestion for emission development. For Delhi, this relation is
mathematically represented in Eq. (3) of Malik et al. (2021). By rearranging, the same can be
written as Eq. (1) of this paper.

$$x_i = c_i \left( \frac{1}{\alpha} \left( \frac{V_{o,i}}{V_{Congested,i}} - 1 \right) \right)^{\frac{1}{\beta}} \tag{1}$$


Where,
$x_i$ = Traffic flow for road link i
$c_i$ = Traffic capacity for road link i
$V_{Congested,i}$ = Speed during congestion (km/h) for link i
$V_{o,i}$ = Free flow velocity (FFV) of traffic for road link i
α and β = constants (Table 1, Malik et al., 2021)

Traffic volume and road capacity determines the traffic speed. Increasing traffic volume leads
to travel time delay (congestion) which further results in road traffic congestion resulting in
increased traffic volume and decreased speed leading to traffic delays. Congested traffic speed
($V_{congested}$) is inversely proportional to the *congestion* (Afrin and Yodo., 2020). Here we define
*congestion* as percentage increase in travel time, i.e. 50% congestion level in a city means that
a trip will take 50% more time than it would during baseline uncongested conditions. In real
world situations, even with the light traffic the congestion exists where minimum time delay is
observed to reduce the likelihood of collision, known as single interaction (Vickrey, 1969).
Therefore, the congestion cannot be zero in large cities such as Delhi with complex urban
geometry and night-time activity. Wei et al. (2022) has reported lowest congestion value raging
from 0.01 to 0.08 during night-time across 77 Chinese cities. In this study, we have used hourly
*congestion* data for Delhi obtained from TomTom (https://www.tomtom.com/en_gb/traffic-
index/about/). TomTom is one of the leading mapping and navigation services providing urban
congestion worldwide. Congestion data has been taken for different days of the week then
combined to create weekdays (Monday to Friday) and weekend (Saturday and Sunday)
profiles. Because FFV (*Vo*) and *congestion* are known for a road link, $V_{congested}$ for weekdays
and weekend has been calculated for each road link using the Eq. (2).

$$V_{congested} = \frac{Vo}{1 + congestion} \tag{2}$$

Further, substituting the value of $V_{congested}$ in Eq. (1), we get a relation between congestion and
traffic flow (Eq. 3) that has been used to estimate the weekdays and weekend traffic flow for
all the road links in personal car units (PCU).

$$x_i = c_i \left( \frac{congestion}{\alpha} \right)^{\frac{1}{\beta}} \qquad congestion > 0 \tag{3}$$

For large cities such as Delhi, the night-time congestion and traffic are not zero. It can be
considered as a smooth traffic flow situation with congestion greater than zero. Therefore, to
avoid zero traffic in equation 3, we have used a minimum congestion value of 0.03 (3%) for
Delhi. We use $c_i$ from TRIPP and *congestion* from TomTom. The values α, β and $c_i$ used in
this study are taken from Malik et al., (2021), and are shown in Table S2. We take three-point
moving average of hourly congestion and calculate the traffic flow using equation 3. The traffic
flow is calculated in terms of PCU. The PCU values for Delhi are taken from Malik et al. (2021)
and are as follows (a) 1.0 for CAR, (b) 0.5 for 2W, (c) 1.0 for 3W, (d) 3.0 for BUS, (e) 1.5 for
LCV and (f) 3.0 for HCV. Malik et al. (2021) has reported speed–volume relationship for
different road classes in Delhi and has given for different lanes (1 lane, 2 lanes, 3 lanes and >4
lanes). In order to harmonize the road classes, we use RClass1 for 1 lane, RClass2 for 2 lanes,
RClass3 for 3 lanes, and RClass4 and RClass5 for >4 lanes. We selected the parameters of the
road classes that have high numbers of sample points and $R^2$ corresponding to each road class.
For e.g., for RClass3, we considered the 3 lanes having higher $R^2$. Further, the speed and traffic
volume has been corrected for each road link to match the observed PCU in TRIPP dataset for
a better agreement. The PCU and speed variation across all road classes are shown as a box
plot in Fig. S5. The comparison of observed and estimated traffic at the 72 location of TRIPP
is shown in Fig. S3. The estimated and measured traffic have a correlation of 0.99 and the
difference (estimated - measured) varies from -0.6% to 2.6%. The hourly estimated traffic for
each road link is further decomposed from PCU to different fleet categories using the
percentage share provided by Malik et al., 2018. The hourly estimated traffic has been further
corrected for the LCV and HCV using the percentage share provided by CRRI (Central Road
Research Institute; Errampalli et al., 2020) to account for the travel restrictions of good vehicles
during peak traffic hours. For simplicity, minibus has been combined with the bus category
and NMVs are not used in this study.  To validate our activity data, the annual VKT estimated
for each fleet category has been compared with earlier reported studies (Sahu et al., 2011;
Kumar et al., 2011; Guttikunda and Calori., 2013; Goel et al., 2015b; Malik et al., 2019) and is
tabulated in Table S11 and discussed in section 3.1.
**2.2 Vehicular Classification:**
The six types of primary vehicle categories (2W, 3W, CAR, BUS, LCV and HCV) have been
further classified into 127 categories (Table S1) according to fuel, engine capacity and emission
standards to match the COPERT-5 vehicular classification. The fuel share of petrol/gasoline,
diesel and CNG/LPG vehicles in Delhi for passenger and freight vehicles has been obtained
from Dhyani and Sharma. (2017) and Malik et al. (2019) respectively. The engine share for
primary vehicle categories has been taken from working papers (Sharpe and Sathiamoorthy.,
2019; Anup and Yang., 2020; Deo and Yang., 2020) of the International Council on Clean
Transportation (ICCT). In India, the emission norms/standards, known as Bharat Stage (BS),
can be considered equivalent to the European Emission Standards - Euro, have been introduced
in a phased manner. These norms were introduced for passenger cars then later extended to
other vehicle categories. For example, the BS-I (India-2000) for passenger cars was
implemented in 2000 followed by BS-II, BS-III and BS-IV in 2005, 2010 and 2017
respectively.  The BS-VI for passenger cars is introduced recently in 2020 therefore has not
been considered in our study.  For Delhi, the timeline of BS implementation for passenger cars
and other vehicles are shown in Table S3. The vehicles prior to the implementation of BS
norms have been considered as Conventional (or BS-0 for simplicity).  The BS share of the
vehicles has been derived using the survival function method described in (Goel et al., 2015b;
Malik et al., 2019). The vehicle survival was calculated for the past twenty years by considering
2018 as the base year and then the BS share was calculated based on the age of the vehicle with
respect to 2018 (Table S4). The final share of the primary vehicle category as per fuel, engine
and BS norms has been calculated by multiplying the fuel share, engine share and BS norms
share and shown in Table S1. In this study, BS and EURO/Euro have been used
interchangeably, and BS-I to BS-IV or BS1 to BS4 or EURO1 to EURO4 represent the same
emission standard.
**2.3 Emission Factors**
Emission factor (EF) is a crucial parameter needed for emission estimation. Road traffic
vehicular emission depends on a variety of factors such as vehicle type, fuel used, engine types,
driving pattern, road type, emission legislation type (BS/EURO) and speed of the vehicle. We
have adopted the recent COPERT-5 tier-3 methodology and used the speed based emission
factor (https://www.emisia.com/utilities/copert/) for 127 vehicle types (Table S1) and
according to the emission legislation up to BS/EURO-4 (As in 2018 BS-VI is not
implemented). The EF as a function of vehicle speed ($v$) is calculated using Eq. (4).

$$\text{EF}(v) = \frac{(\alpha \times v^2) + (\beta \times v) + \gamma + \left(\frac{\delta}{v}\right)}{(\varepsilon \times v^2) + (\zeta \times v) + \eta} \tag{4}$$

Where,
v is the speed,
$\alpha$, $\beta$, $\gamma$, $\delta$, $\varepsilon$, $\zeta$ and $\eta$ are coefficients that varies with vehicle type

The coefficients for each pollutant and vehicle category are taken from the COPERT-5
database (COPERT-5 Guidebook, 2020). The emission factors are further corrected for the
emission degradation occurring in older vehicles considering the mileage as discussed in
(COPERT-5 Guidebook, 2020). COPERT relies on mean driving speed and travel distance.
The mean speeds are relatively low under urban driving conditions, and emission factors are
highly variable within this speed range due to the speed fluctuations caused due to real-time
driving behaviour (frequent braking, acceleration, deceleration, idling). Lejri et al. (2018) have
estimated the relative errors on fuel consumption and $NO_x$ emissions related to mean speed
variations from 2 to 10 km/h and estimated errors up to 25-30% in fuel consumption and $NO_x$
emissions. Therefore, to account for the emissions due to the speed fluctuations around the
mean speed, a factor of 1.2, i.e. 20% increase has been applied to the final dataset. This has
been applied for all the hours and all the pollutants. Although we apply the same factor for all
hours of the day, the added emissions are more during high congestion hours and less during
low congestion hours.
The non-exhaust emissions (Singh et al., 2020) have not been calculated in this study. As
COPERT does not provide the EFs for the 3W CNG category, we have used EFs of CNG mini
CAR for this. BC and OM emission are computed using the fraction (by COPERT-5
Guidebook, 2020) from PM exhaust. We have compared the COPERT EFs used in this study
with the earlier reported EFs and shown in Table S12 to elaborate upon the potential uncertainty
in the key vehicle categories. Further, the emission uncertainties have been discussed in section

308    4.

**2.4 Emission calculation**
The model calculates hourly emissions for each road link of finite length and uses hourly traffic
volume and emission factors as a function of speed for 127 vehicle categories (Table S1). The
hourly emission rate ($Q$) for each road link is calculated using Eq. (5). The total emission for a
given hour is calculated by taking the sum of emission across all vehicle categories.

$$Q_{i,h}^{p} = \sum_{j} V_{i,j,h} \times EF_{j}^{p}(v_{i,h}) \times L_{i} \tag{5}$$

Where
$Q_{i,h}^{p}$ is emission rate of a pollutant $p$ for road link $i$ and at hour $h$, where $h=0$ to $23$
$V_{i,j,h}$ is the traffic volume of vehicle category j for road link i at hour h, where $j=1$ to $127$
$L_{i}$ is the length of road link i
$EF_{j}^{p}(v_{i,h})$ is the emission factor of pollutant $p$ for vehicle category $j$ as a function speed $v_{i,h}$
for road link $i$ at hour $h$.
The hourly emissions have been calculated for each pollutant over each road link then gridded
at 100 m × 100 m resolution using the methodology described in Singh et al., (2018, 2020) to
produce the hourly gridded emission inventory for Delhi.

## 3 Results

### 3.1 Diurnal variation of traffic volume and speed

The estimated hourly traffic volume (in PCU) and speed profiles for Delhi are shown in Fig. 2. An anticorrelated diurnal variation is seen in the traffic volume and speed. The weekdays traffic volume tends to have a bimodal profile with a morning peak (09:00-11:00) and an evening peak (18:00-20:00). A similar traffic volume profile has also been observed by other studies over Delhi (Dhyani and Sharma., 2017; Sharma et al., 2019). Similar bimodal traffic profile is also observed over the cities around the world subject to the city specific travel demand (Järvi et al., 2008 for Helsinki; Jing et al., 2016 for Beijing) The evening peak traffic volume tends to be 40% higher than the morning peak. The vehicular composition changes hourly (Fig. S1) and also varies with respect to the road classes (Table S5). The night-time goods vehicle share is more in comparison to the passenger and personal vehicles (Fig. S1). The weekend traffic volume does not show a morning peak due to closure of the offices/workplaces and shows evening peaks due to shopping and other weekend activities. As usual the minimum traffic volume is observed at night (00:00-04:00 hours) because of the reduced human and commercial activities. Due to the minimum traffic at night, the traffic moves with an average speed of $51\pm6$ km/h with almost no congestion. As traffic volume increases, it starts to build congestion, leading to reduced speed. The average speed during the weekdays morning peak hours is estimated to be $30\pm14$ km/h whereas the evening speed is estimated to be $28\pm15$ km/h. The evening congestion leads to an average 46% reduction in the average speed increasing the travel time by a factor of two. We calculated the average profiles for each road link by combining weekdays and weekends and used them in the emission calculations. The estimated profiles averaged across all road links are shown in Fig. 2.

We have estimated 27, 31, 6. 1.7, 0.95 and 3.14 billion VKT driven by CAR, 2W, 3W, BUS, HCV and LCV respectively. The comparison between estimated annual VKT and reported by other studies is tabulated in TableS11. This comparison table includes the studies which have either reported annual VKT or have provided enough data to calculate annual VKT. The VKT values compare well with the earlier studies by considering the fact that the uncertainties exist in the method of estimation, year and study domain. Malik et al. (2019) estimated the destined and non-destined VKT of freight vehicles (HCV and LCV) with the actual measured traffic at several entry points in Delhi. Goel et al. (2015b) estimated the annual VKT based on the annual mileage of the 2W and cars obtained from PUC (Pollution under control) certification data and the number of registered vehicles. The VKT reported by Goel et al. (2015b) for Cars and 2W

are slightly lower than our study. The study by Goel et al. was conducted in 2012 since then
the cars and taxis share has almost doubled in Delhi due to increased travel demand and
economic growth (DDA, 2021). The study by Kumar et al. (2011), which is for 2010, reported
higher VKT for Buses and HCV as compared to the one estimated by the current study. Their
estimates were based on the assumed distance travelled by each vehicle and the number of
registered vehicles than the actual on road vehicle. Guttikunda and Calori. (2013) reported
high VKT for buses and HCV. The study by Sahu et al. (2011) for NCR Delhi estimated very
high VKT for 2W and Cars. While earlier studies have reported different VKT values the
relative VKT share compares well with our study. Moreover, the VKT estimated by recent
studies are close to our estimates.

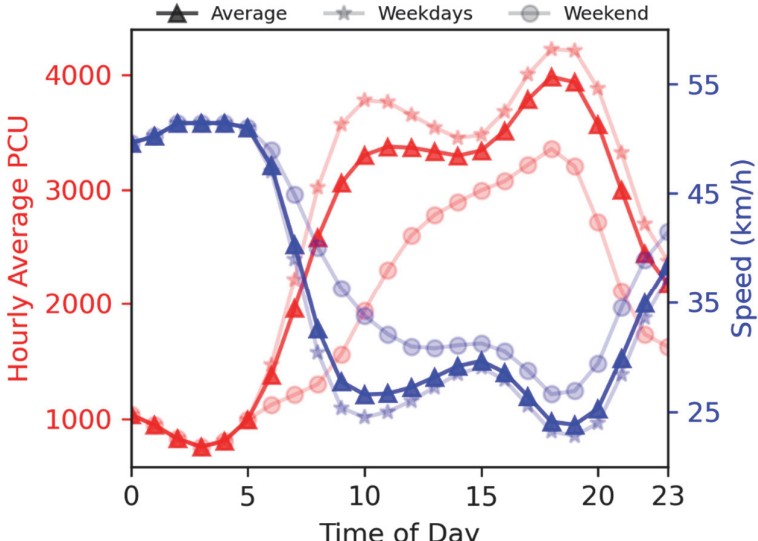


Figure 2. Weekdays, weekend and average diurnal profile for traffic volume in average PCU
(red) and average speed (blue) over Delhi. The legend reflects the different markers used for
weekdays, weekend and average profile.
**3.2 Emission inventory**
A multi-pollutant hourly and high spatial resolution (100m × 100m) emission inventory has
been prepared for Delhi. As an example, the spatial distribution of $NO_x$ emission at 03:00-
04:00, 09:00-10:00, 15:00-16:00 and 18:00-19:00 hours, representing early morning, morning
peak, afternoon and evening peak respectively, has been shown in Fig. 2. The emission rate
during the evening peak hours is the highest during the day followed by morning peak hours.
The high traffic volume along with traffic congestions lead to more emissions during the peak
traffic hours (Jing et al., 2016). The emission during the afternoon hours is comparable or less
than that of the morning hours whereas the early morning emissions are lowest because of low
traffic volume moving with free flow speed. The diurnal profile of emissions has been
discussed in detail in Section 3.5.
The annual emissions have been calculated by summing the hourly emissions to get daily
emissions and then multiplying with 365 (number of days in a year) to get annual emissions.
The monthly variation in the emission has not been considered as the monthly variations are
much smaller than the hourly variations. We estimated an annual emission of 1.82 Gg for PM,
0.94 Gg for BC, 0.75 Gg for OM, 221 Gg for CO, 56 Gg for $NO_x$, 64 Gg for VOC, 0.28 Gg for
$NH_3$, 0.26 Gg for $N_2O$ and 11.38 Gg for $CH_4$ in 2018.

**3.3 Spatial variation**
The hourly emissions over Delhi have been summed together to calculate the daily emissions
for all the pollutants. The spatial variation of daily mean emission rate has been analysed over
three selected regions, viz. inner, outer and eastside Delhi (as shown in Fig. 1). The total
emission for each pollutant and for each region has been tabulated in Table S6. Outer Delhi
region has the highest emission (51-53%) for all the pollutants because of its largest area of
1106 $km^2$ which is 4.5 times of inner Delhi. To avoid the influence of area on the emissions,
we have calculated the emission flux (i.e. emission per unit area) and shown in Table S7. The
emissions flux is highest for inner Delhi followed by eastside and outer Delhi region. For all
pollutants, the emissions flux in inner Delhi is 40 - 50 % higher than the average emission of
Delhi whereas the emissions flux in outer Delhi is ~46% lower. The emission flux is
consistently high along the grids containing major roads (Fig. 3), intersections and major
business hubs. Inner Delhi consists of major business hubs, workplaces and government
offices, which entertain more vehicular activity in this region resulting in congestion leading
to reduced speed and enhanced emissions. The daytime average speed across all roads in Inner
Delhi is 29 km/h which is lower than the daytime average speed of 32 km/h in outer Delhi. The
lower speed and higher traffic density influences the economic driving behaviour resulting in
frequent braking, idling, acceleration and deceleration that enhances the vehicular emission.
Moreover, the morning and evening peak hours with higher traffic and lower speed have the
highest emission as compared to the rest of the day. In these heavy congested hours, the vehicle
is forced to run in lower speed which boosts the emission.

**3.4 Emissions along the Road class**
The emissions along the five road classes used in this study have been calculated and shown in
Table 1 and the hourly variation of emission has been shown in Fig. 4. RClass3 has a
substantial emission share (~35%) across all pollutants followed by RClass5 and RClass2,
whereas RClass1 holds the minimum emissions share (~2-3%). The dominant emission share
of RClass3 is due to the optimum vehicular activities over the longer road length. RClass2,
which are the feeder roads to the RClass3, RClass4 and RClass5, contribute ~23% to the
emission. The multi-lane wider roads, RClass4 and RClass5 contribute ~13-15 % and ~21-25
% respectively to the total emission. To remove the dependency of the road length, we
calculated the emission per km segment of a road. The emissions (per km) over multi-lane
wider roads (RClass4 and RClass5) are almost two times of the RClass3 (Table S8 and Fig.
S2) due to more traffic flow irrespective of the congested conditions. However, the emission
per lane per kilometre (Table S9) for RClass1 is found to be the highest because of lower speed
and congestion and major share of 2W. This shows that effective management of traffic in
narrow roads to reduce the congestion will be beneficial in reducing the pollution without
impacting the traffic volume. The multi-lane wider roads (RClass4 and RClass5) help the
vehicle to maintain an economic speed resulting in minimum congestion and lower emission,
however they are the emission hotpots in Delhi.
Table 1. Emission in Mega gram (Mg) per day (% share) across different road types.

| RClass | PM | BC | OM | CO | $NO_x$ | VOC | $NH_3$ | $N_2O$ | $CH_4$ |
|--------|-----|------|------|-----|------|------|------|------|-------|
| **RClass1** | 0.16 (3%) | 0.09 (3%) | 0.07 (3%) | 19 (3%) | 4 (2%) | 5 (2%) | 0.02 (2%) | 0.02 (2%) | 1.0 (3%) |
| **RClass2** | 1.17 (23%) | 0.61 (23%) | 0.49 (23%) | 139 (23%) | 35 (23%) | 41 (23%) | 0.16 (21%) | 0.16 (22%) | 7.3 (23%) |
| **RClass3** | 1.77 (35%) | 0.9 (34%) | 0.75 (36%) | 228 (37%) | 52 (34%) | 67 (38%) | 0.27 (35%) | 0.25 (35%) | 11.29 (36%) |
| **RClass4** | 0.72 (14%) | 0.38 (14%) | 0.29 (14%) | 84 (13%) | 22 (14%) | 23 (13%) | 0.12 (15%) | 0.11 (15%) | 4.43 (14%) |
| **RClass5** | 1.16 (23%) | 0.62 (23%) | 0.46 (22%) | 132 (21%) | 38 (25%) | 37 (21%) | 0.19 (25%) | 0.17 (23%) | 7.19 (23%) |


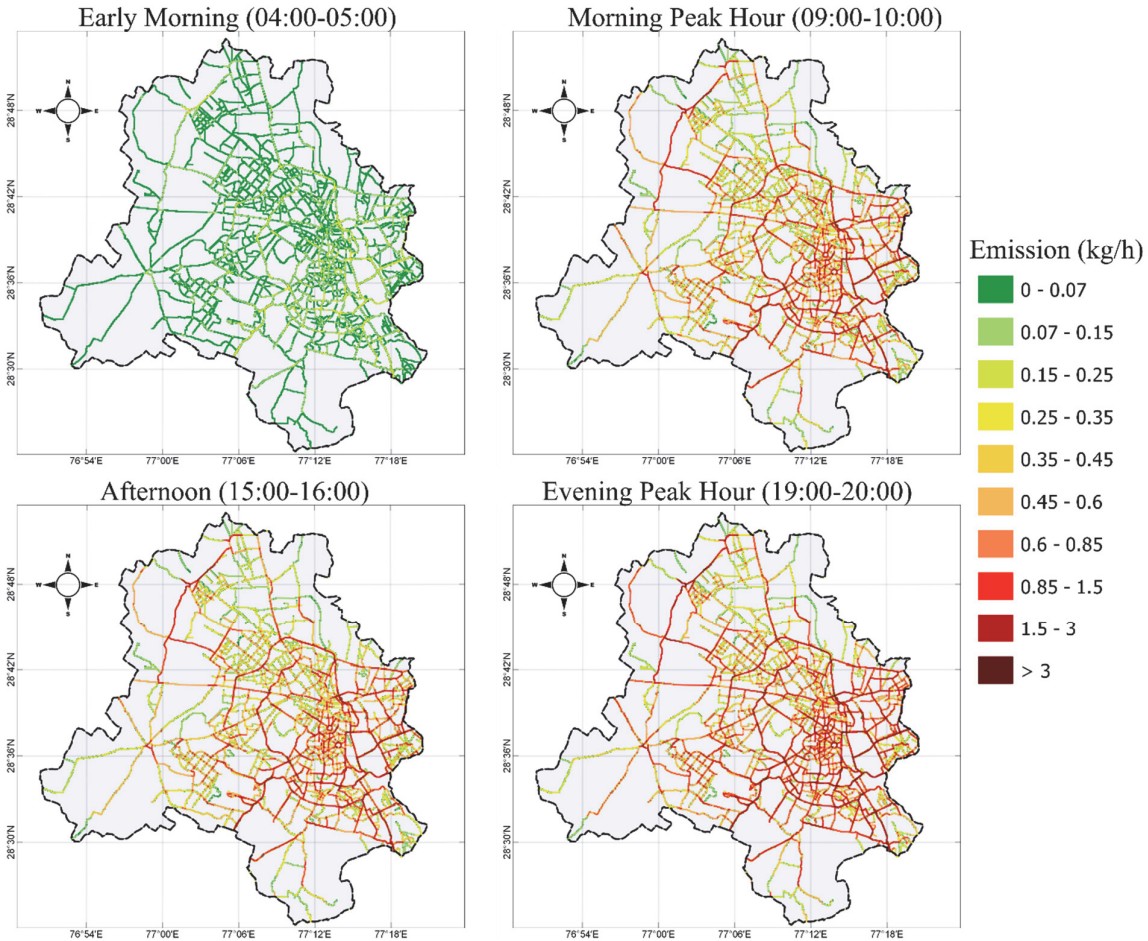


Figure 3. Estimated gridded NO$_x$ emission in kg/h (kilogram per hour) at 100m × 100m
spatial resolution at different times of the day representative of different congestion levels.

**3.5 Diurnal variation of emission**
Dynamic traffic volume and speed, as discussed in section 3.1, results in diurnal variation in
the emissions during a day. Fig. 4 shows the hourly emissions (Mg/h) and contribution of each
road class at each hour in Delhi. The temporal evolution of emission is linear with the traffic
variation in a day with the minimum variation during the night-time and remarkable variation
during the human active hours (08:00-20:00). Among different road types and for all the
pollutants RClass1 has the lowest and RClass3 has the highest emission proportional to the
traffic volume. A similar temporal variation of NO$_x$ emission rate is observed in a study, for
different road types of Beijing (Jing et al., 2016). For most of the pollutants (except PM, BC
and NO$_x$), daytime (08:00 to 20:00) contributes ~70% to the daily emissions whereas the
morning (09:00 to 11:00) and evening (18:00 to 20:00) rush hours alone altogether add 30-40%
to the total emissions. The increasing activity of goods vehicles (HCV + LCV) during afternoon
and night-time (Fig. S1) elevates the emission of PM, BC and $NO_x$ from these vehicles (Fig.
5) resulting in a different diurnal profile compared to other pollutants. The $NO_x$ and particulate
pollutants (PM and BC) emissions during late night hours (11:00-05:00) is relatively higher,
adding up to 60% and 75% of total particulate and $NO_x$ night-time emissions respectively as
shown in Fig. 5. The contribution of vehicle type has been discussed in detail in section 3.6.
The diurnal evolution of emission is also visible in the hourly spatial map shown in Fig. 3.
Early morning with minimum traffic volume has lower emission whereas the evening rush hour
with increasing congestion has higher emission. The density of higher emission grids (Fig. 3)
in the inner Delhi region is higher compared to other regions throughout the day.

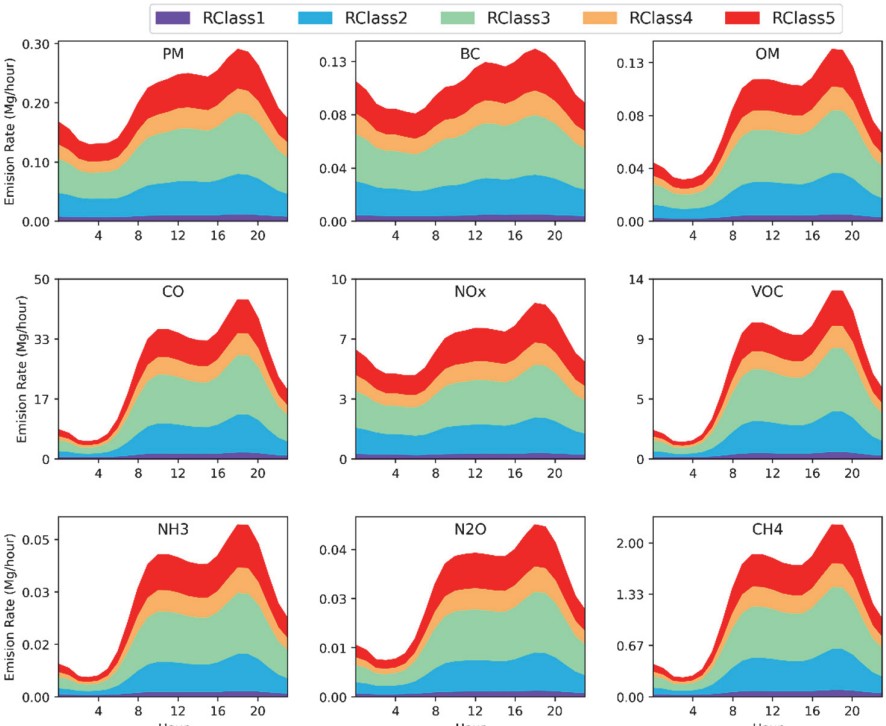

Figure 4. Variation of hourly emission (in megagram/hour) of the nine pollutants averaged
across Delhi according to the five road classes (RClass1 to RClass5). Different colors
indicate the hourly contribution of each RClass to the total emission.

**3.6 Vehicular emission share**
The percentage share of major vehicle types to the total emission of nine pollutants has been
calculated and shown in Table 2 and its hourly contribution is shown in Fig. 5. The 2W
vehicles, having a major vehicular share (Table S5), are the major contributors to the total
emissions for all the pollutants except for BC, $NO_x$ and $N_2O$. The goods vehicles (HCV and
LCV) contribute substantially, mainly during night-time, to the PM, BC and NO$_x$ emissions.
Buses have the highest contribution to NO$_x$ emissions and substantial contribution to PM, BC
and CH$_4$. Cars are the dominant source for NH$_3$ and N$_2$O and contribute substantially to PM,
BC and NO$_x$ emissions. However, most of the emissions are from diesel cars.

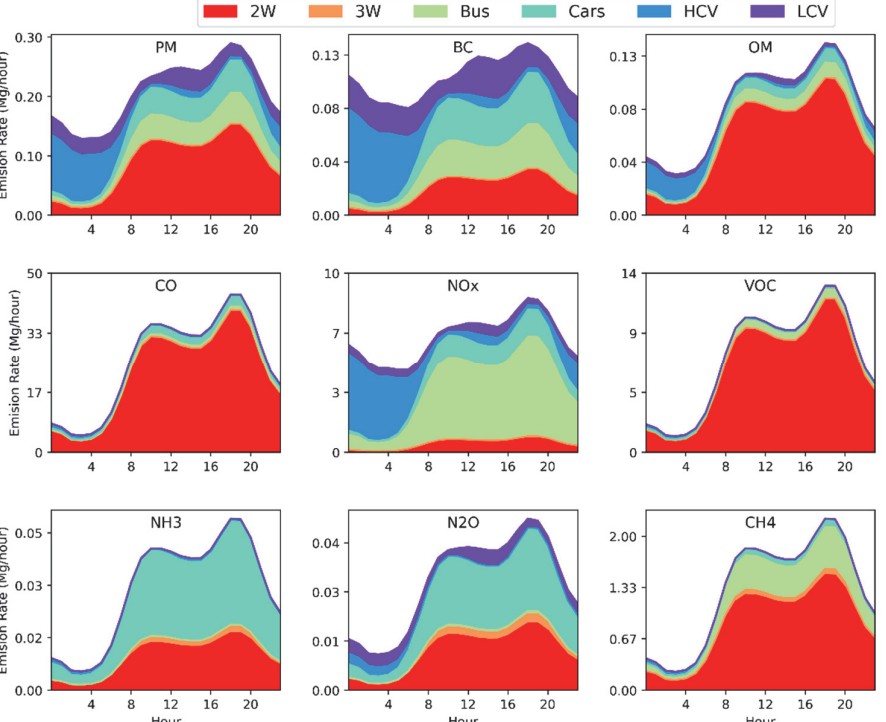


Figure 5. Variation of hourly emission (megagram/hour) of the nine pollutants averaged
across Delhi according to the major vehicle type.  Different colors indicate the hourly
contribution of each vehicle type to the total emission.

Table 2. Emission in kg/day (% share) according to the vehicle types.

| Vehicle | PM | BC | OM | CO | NO$_x$ | VOC | NH$_3$ | N$_2$O | CH$_4$ |
|---------|-----|-----|-----|-----|--------|-----|--------|--------|--------|
| 2W | 2102 (41.6%) | 500 (19.0%) | 1475 (71.5%) | 532316 (88.0%) | 10600 (6.8%) | 159582 (90.5%) | 249 (32.6%) | 249 (35.4%) | 20588 (66.0%) |
| Cars | 740 (14.6%) | 537 (20.4%) | 146 (7.1%) | 42276 (7.0%) | 20185 (12.9%) | 3546 (2.0%) | 458 (60.0%) | 308 (43.8%) | 1425 (4.6%) |
| 3w | 25 (0.5%) | 3 (0.1%) | 11 (0.5%) | 3305 (0.5%) | 1593 (1.0%) | 952 (0.5%) | 32 (4.2%) | 35 (5.0%) | 1151 (3.7%) |
| Buses | 691 (13.7%) | 459 (17.4%) | 160 (7.8%) | 12739 (2.1%) | 75536 (48.4%) | 9249 (5.2%) | 4 (0.5%) | 12 (1.7%) | 7456 (23.9%) |
| HCV | 787 (15.8%) | 546 (21.2%) | 171 (8.3%) | 8645 (1.4%) | 35404 (23.0%) | 2057 (1.2%) | 9 (1.2%) | 24 (3.4%) | 452 (1.4%) |
| LCV | 636 (12.8%) | 534 (20.7%) | 87 (4.2%) | 4803 (0.8%) | 10547 (6.9%) | 884 (0.5%) | 11 (1.4%) | 75 (10.7%) | 126 (0.4%) |


Table 3. Emission in kg/day (% share) according to fuel type.

| Fuel | PM | BC | OM | CO | NO$_x$ | VOC | NH$_3$ | N$_2$O | CH$_4$ |
|---|---|---|---|---|---|---|---|---|---|
| CNG | 95 (1.9%) | 14 (0.5%) | 43 (2.1%) | 12703 (2.1%) | 45832 (29.8%) | 9335 (5.3%) | 68 (8.9%) | 73 (10.4%) | 9547 (30.6%) |
| Diesel | 2698 (54.1%) | 2052 (79.5%) | 491 (23.9%) | 25583 (4.2%) | 91144 (59.2%) | 5308 (3.0%) | 36 (4.7%) | 225 (32.0%) | 805 (2.6%) |
| Petrol | 2191 (44.0%) | 514 (19.9%) | 1517 (74.0%) | 565799 (93.7%) | 16890 (11.0%) | 161628 (91.7%) | 662 (86.4%) | 406 (57.7%) | 20848 (66.8%) |


The vehicular fuel share to the total emission for each pollutant is shown in Table 3. Petrol
vehicles are the largest contributors to the CO (~94%), VOC (91%), NH$_3$ (86%), OM (74%),
CH$_4$ (67%) and N$_2$O (58%) whereas diesel vehicles are the largest contributor to the BC
(~80%), NO$_x$ (59%) and PM (54%) emissions. The contribution of the CNG vehicles is
relatively smaller except for the NO$_x$ and CH$_4$ where they contribute to ~30 %, almost one
third, to the total emissions.

The larger contribution of petrol to the VOC, CO, OM and CH$_4$ emissions are dominated by
2W where we estimated that 2W in Delhi alone contribute 90%, 88%, 71%, and 66%
respectively as shown in Table 2.  The contribution of 2W is also highest to PM (42%). The
larger share of 2W towards the CO emissions has also been reported earlier, 61% in Goyal et
al., (2013); 43% in Sharma et al., (2016) and 37% in Singh et al., (2018).  Higher emission
share of 2W is due the higher emission factor of VOC in petrol fuelled 2W (Hakkim et al.,
2021) that has been also reported in a multi-year emission study over Delhi by Goel et al.
(2015a).

The PM emissions are dominated by diesel fuelled HCVs (16 %), LCVs (13%), Buses (14 %)
and Cars (~13 %), whereas 2W are the main source in petrol fuelled vehicles contributing ~42%
to the total PM emissions. Earlier, Sharma et al. (2016) reported 33% share of 2W emission in
2014.  The share of petrol cars and CNG buses towards the PM, BC and OM emissions is less
than 2%. While it is clear that diesel powered vehicles are the major source of PM emission,
earlier studies have reported similar results but with large variations of HCVs in emission share.
The largest share of diesel fuelled HCV is reported as 92% by Goyal et al. (2013), 46% by
Sharma et al. (2016) and 33% by Singh et al. (2018). All these studies reported minimal
emission share (less than 10% combining both diesel and petrol cars). The largest share of
HCV, LCV and diesel Cars to BC emission is because of higher emission factors (Zavala et
al., 2017) contributing to total urban BC emission as shown by Bond et al., (2013).

The petrol cars contribute more than half of the total $NH_3$ emissions and among them the Euro
2 with higher emission factor has the largest share of 39%. The diesel vehicles (HCVs, LCVs,
diesel Buses and Cars) altogether contribute significantly to the PM, BC and $NO_x$ emissions.
The higher emission factor of diesel fuelled vehicles (Wu et al., 2012) clearly reflects in the
emission share.

CNG buses have the highest share (27%) in $NO_x$ emission and around 23% in $CH_4$ emissions.
The highest share of CNG is due to higher $NO_x$ emission factor for CNG vehicles compared to
petrol vehicles (Dimaratos et al., 2019). The larger share of ~15% from CNG buses to the total
traffic $NO_x$ emission is also reported in a study of CPCB (2010).  In terms of Euro or BS
standard, Euro 3 vehicles have the highest share (Table S10) in the total emission except for
$N_2O$ and $NH_3$. This is mainly because of the highest share of Euro 3 vehicles in 2W, Buses,
HCV and LCV (Table S4 in the Supplement). In the case of $N_2O$, the emissions are dominated
by Euro 4 cars which have around 84% share to the total cars. For $CH_4$, the highest share of
Euro 3 vehicles is due to the higher emissions from Euro 3  2W as the emission factor of petrol
vehicles is higher (Clairotte et al., 2020).

In order to have a clear picture of the dominant polluting vehicle categories, we grouped
different vehicle types into 35 categories and calculated the percentage share to the total
emission of nine pollutants as shown in Table 4. We further identified the top five polluting
vehicle categories for each pollutant and tabulated in Table 5.  For PM, the top five polluting
vehicles account for 55% of the total emissions which is dominated by petrol Euro 3 petrol
2W and Euro 3 diesel HCVs. The BC emission is mainly driven by Euro 3 diesel HCVs, LCVs,
Buses and the top five polluting vehicles account for 66% of the total emissions. The OM, CO,
VOC emissions are dominated by 2W and the top five accounts for 71%, 89% and 91% of total
emissions respectively.

Petrol fuelled cars and 2W hold the dominant share of $NH_3$ emissions because of the larger EF
compared to other categories (COPERT-5 Guidebook, 2020). For $N_2O$, 2W Euro 3 holds the
highest share of 21%, followed by EURO IV diesel and petrol cars. The top five contributors
to $CH_4$ emissions account for 86% of the total emissions which are dominated by 2W and CNG
buses. These two categories of vehicles altogether contribute to ~97% of the emissions.

Table 4. Emission share of vehicles of different class, fuel and BS/EURO standards.
Contributions less than 0.1% are not shown here. Contributions more than 10% are shown in
the same colour. (D: Diesel, P: Petrol, C: CNG and number 0-4 represents the Euro type starting
from 0 being conventional to 4 as Euro 4).

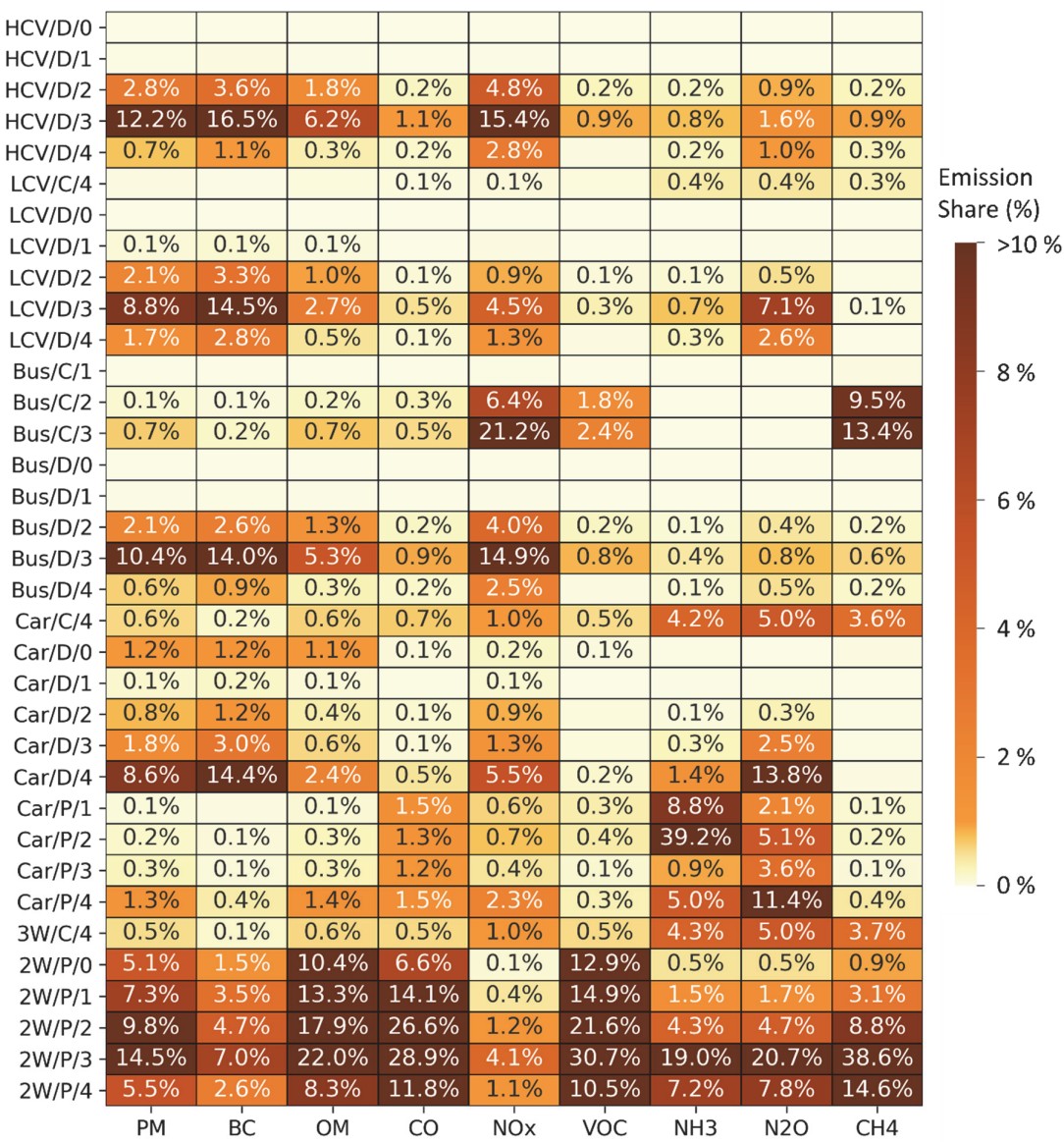

| | PM | BC | OM | CO | NOx | VOC | NH3 | N2O | CH4 |
|---|---|---|---|---|---|---|---|---|---|
| HCV/D/0 | | | | | | | | | |
| HCV/D/1 | | | | | | | | | |
| HCV/D/2 | 2.8% | 3.6% | 1.8% | 0.2% | 4.8% | 0.2% | 0.2% | 0.9% | 0.2% |
| HCV/D/3 | 12.2% | 16.5% | 6.2% | 1.1% | 15.4% | 0.9% | 0.8% | 1.6% | 0.9% |
| HCV/D/4 | 0.7% | 1.1% | 0.3% | 0.2% | 2.8% | | 0.2% | 1.0% | 0.3% |
| LCV/C/4 | | | | 0.1% | 0.1% | | 0.4% | 0.4% | 0.3% |
| LCV/D/0 | | | | | | | | | |
| LCV/D/1 | 0.1% | 0.1% | 0.1% | | | | | | |
| LCV/D/2 | 2.1% | 3.3% | 1.0% | 0.1% | 0.9% | 0.1% | 0.1% | 0.5% | |
| LCV/D/3 | 8.8% | 14.5% | 2.7% | 0.5% | 4.5% | 0.3% | 0.7% | 7.1% | 0.1% |
| LCV/D/4 | 1.7% | 2.8% | 0.5% | 0.1% | 1.3% | | 0.3% | 2.6% | |
| Bus/C/1 | | | | | | | | | |
| Bus/C/2 | 0.1% | 0.1% | 0.2% | 0.3% | 6.4% | 1.8% | | | 9.5% |
| Bus/C/3 | 0.7% | 0.2% | 0.7% | 0.5% | 21.2% | 2.4% | | | 13.4% |
| Bus/D/0 | | | | | | | | | |
| Bus/D/1 | | | | | | | | | |
| Bus/D/2 | 2.1% | 2.6% | 1.3% | 0.2% | 4.0% | 0.2% | 0.1% | 0.4% | 0.2% |
| Bus/D/3 | 10.4% | 14.0% | 5.3% | 0.9% | 14.9% | 0.8% | 0.4% | 0.8% | 0.6% |
| Bus/D/4 | 0.6% | 0.9% | 0.3% | 0.2% | 2.5% | | 0.1% | 0.5% | 0.2% |
| Car/C/4 | 0.6% | 0.2% | 0.6% | 0.7% | 1.0% | 0.5% | 4.2% | 5.0% | 3.6% |
| Car/D/0 | 1.2% | 1.2% | 1.1% | 0.1% | 0.2% | 0.1% | | | |
| Car/D/1 | 0.1% | 0.2% | 0.1% | | 0.1% | | | | |
| Car/D/2 | 0.8% | 1.2% | 0.4% | 0.1% | 0.9% | | 0.1% | 0.3% | |
| Car/D/3 | 1.8% | 3.0% | 0.6% | 0.1% | 1.3% | | 0.3% | 2.5% | |
| Car/D/4 | 8.6% | 14.4% | 2.4% | 0.5% | 5.5% | 0.2% | 1.4% | 13.8% | |
| Car/P/1 | 0.1% | | 0.1% | 1.5% | 0.6% | 0.3% | 8.8% | 2.1% | 0.1% |
| Car/P/2 | 0.2% | 0.1% | 0.3% | 1.3% | 0.7% | 0.4% | 39.2% | 5.1% | 0.2% |
| Car/P/3 | 0.3% | 0.1% | 0.3% | 1.2% | 0.4% | 0.1% | 0.9% | 3.6% | 0.1% |
| Car/P/4 | 1.3% | 0.4% | 1.4% | 1.5% | 2.3% | 0.3% | 5.0% | 11.4% | 0.4% |
| 3W/C/4 | 0.5% | 0.1% | 0.6% | 0.5% | 1.0% | 0.5% | 4.3% | 5.0% | 3.7% |
| 2W/P/0 | 5.1% | 1.5% | 10.4% | 6.6% | 0.1% | 12.9% | 0.5% | 0.5% | 0.9% |
| 2W/P/1 | 7.3% | 3.5% | 13.3% | 14.1% | 0.4% | 14.9% | 1.5% | 1.7% | 3.1% |
| 2W/P/2 | 9.8% | 4.7% | 17.9% | 26.6% | 1.2% | 21.6% | 4.3% | 4.7% | 8.8% |
| 2W/P/3 | 14.5% | 7.0% | 22.0% | 28.9% | 4.1% | 30.7% | 19.0% | 20.7% | 38.6% |
| 2W/P/4 | 5.5% | 2.6% | 8.3% | 11.8% | 1.1% | 10.5% | 7.2% | 7.8% | 14.6% |

Emission Share (%)
>10 %
8 %
6 %
4 %
2 %
0 %


Table 5. Top five polluting vehicle categories for each pollutant.

| PM | BC | OM |
|---|---|---|
| Top 5 accounts for **55%** emissions<br>1. 14% from 2W (Petrol, Euro 3)<br>2. 12% from HCV (Diesel, Euro 3)<br>3. 10% from Bus (Diesel, Euro 3)<br>4. 10% from 2W (Petrol Euro 2)<br>5. 9% from LCV (Diesel Euro 3) | Top 5 accounts for **66%** emissions<br>1. 17% from HCV (Diesel Euro 3)<br>2. 14% from LCV (Diesel Euro 3)<br>3. 14% from Car (Diesel Euro 4)<br>4. 14% from Bus (Diesel Euro 3)<br>5. 7% from 2W (Petrol Euro 3) | Top 5 accounts for **71%** emissions<br>1. 22% from 2W (Petrol, Euro 3)<br>2. 18% from 2W (Petrol, Euro 2)<br>3. 13% from 2W (Petrol, Euro 1)<br>4. 10% from 2W (Petrol, Euro 0)<br>5. 8% from 2W (Petrol, Euro 4) |
| **CO** | **NO$_x$** | **VOC** |
| Top 5 accounts for **89%** emissions<br>1. 29% from 2W (Petrol, Euro 3)<br>2. 27% from 2W (Petrol, Euro 2)<br>3. 14% from 2W (Petrol, Euro 1)<br>4. 12% from 2W (Petrol, Euro 4)<br>5. 7% from 2W (Petrol, Euro 0) | Top 5 accounts for **63%** emissions<br>1. 21% from Bus (CNG, Euro 3)<br>2. 15% from HCV (Diesel, Euro 3)<br>3. 15% from Bus (Diesel, Euro 3)<br>4. 6% from Bus (CNG, Euro 2)<br>5. 6% from Car (Diesel Euro 4) | Top 5 accounts for **91%** emissions<br>1. 31% from 2W (Petrol, Euro 3)<br>2. 22% from 2W (Petrol, Euro 2)<br>3. 15% from 2W (Petrol, Euro 1)<br>4. 13% from 2W (Petrol, Euro 0)<br>5. 10% from 2W (Petrol, Euro 4) |
| **NH$_3$** | **N$_2$O** | **CH$_4$** |
| Top 5 accounts for **79%** emissions<br>1. 39% from Car (Petrol, Euro2)<br>2. 19% from 2W (Petrol, Euro3)<br>3. 9% from Car (Petrol, Euro1)<br>4. 7% from 2W (Petrol, Euro4)<br>5. 5% from Car (Petrol, Euro4) | Top 5 accounts for **61%** emissions<br>1. 21% from 2W (Petrol, Euro 3)<br>2. 14% from Car (Diesel, Euro 4)<br>3. 11% from Car (Petrol, Euro 4)<br>4. 8% from 2W (Petrol, Euro 4)<br>5. 7% from LCV (Diesel, Euro 3) | Top 5 accounts for **86%** emissions<br>1. 39% from 2W (Petrol, Euro 3)<br>2. 15% from 2W (Petrol, Euro 4)<br>3. 13% from Bus (CNG, Euro 3)<br>4. 10% from Bus (CNG, Euro 2)<br>5. 9% from 2W (Petrol, Euro 2) |


## 4 Uncertainty in emissions:

The emission uncertainty depends on the uncertainty of the model internal parameters (e.g.
emission factors) and the uncertainty of the external parameters or input data (e.g. traffic
activity, i.e. traffic volume and speed, distance travelled, vehicle category share, engine share,
fuel share, technology share etc.). Emissions are also influenced by environmental factors such
as relative humidity, temperature (Kouridiset al., 2010; Dey et al., 2019). In most cases, model
outputs are contingent on the accuracy of the input data. Because of the lack of very detailed
spatio-temporal activity data, the calculated emissions are highly uncertain.
We have made an attempt to estimate the uncertainty in emissions of CO, PM, NOx and VOC
for which speed-based emission factors are available. We have calculated the uncertainty in
the emissions by performing sensitivity analysis to VKT and EF. VKT is a good proxy to
represent the traffic activity. First, we have estimated the uncertainty of ~40% and ~80% in
VKT and EF respectively based on the reported VKT and EF by earlier studies as shown in
Table S11 and Table S12 respectively. Then we have calculated the total emission of pollutants
by varying the VKT from -40% to +40% of the VKT estimated by our study and by varying
the EF from -80% to +80% with an interval of 10%. The obtained distribution of the emission
of pollutants is shown in Fig. 6.  We calculated the coefficient of variation (CoV = [Std/
Mean]*100%) of the distribution and estimated an uncertainty of 61%, 60%, 63% and 68% for
CO, PM, $NO_x$ and VOC respectively. Dey et al., (2019) had estimated uncertainties of the
emission of CO, VOC and NMVOC for Ireland in the range of −58% to +76%. Kouridis et al.
(2010) estimated coefficient of variation of 10% for $CO_2$, in the order of 20-30% for $NO_x$,
VOC, $PM_{2.5}$, $PM_{10}$, 50-60% for CO and $CH_4$ and over 100% for $N_2O$.

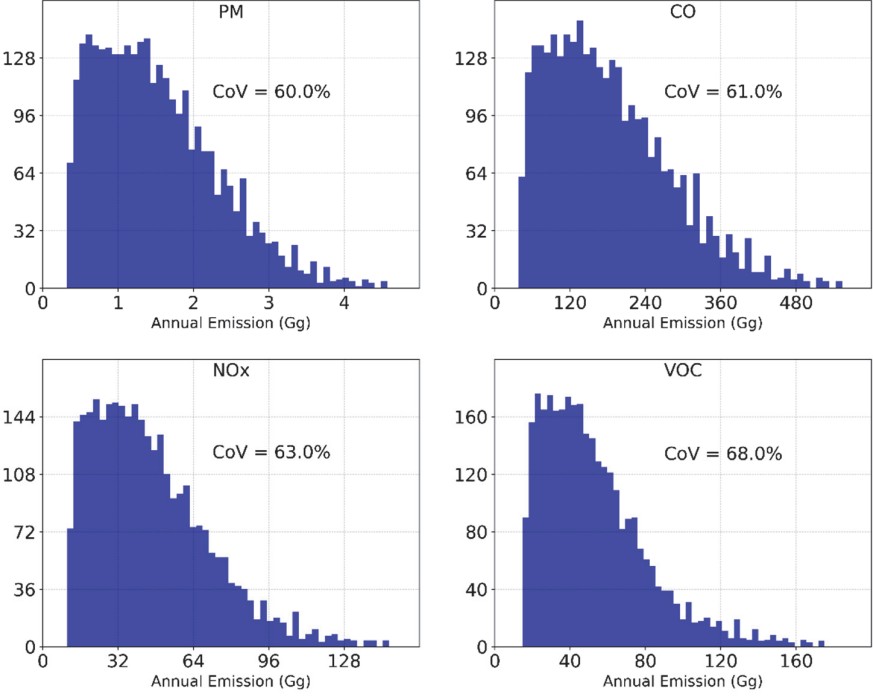


Figure 6. Histogram showing the variation in the annual emissions with the combination of
sensitive parameters (VKT and EF).

**5 Limitations**:
Geotagged dynamic traffic information and emission factors are the backbone of the emission
inventory model. The traffic volume information is very crucial and traditionally obtained by
manual counting or automated counters or through video surveillance at a few locations.
However, in a real-world scenario, the traffic volume and speed can have large variations
within a segment of a road. In this study we have adopted the congestion based approach (Jing
et al., 2016; Yang et al., 2019) to model the traffic volume for each hour of the day. We use
the same diurnal congestion profiles for all roads that could lead to emission uncertainty (Malik
et al., 2021). In reality, some of the roads can be more congested than other roads based on the
local population and traffic management.
The fleet composition can be different for different locations and at a given time of the day
(Sharma et al., 2019). We have used the fleet composition based on surveyed composition at
72 locations during the daytime (08:00-14:00) (TRIPP). To account for the peak hour and day-
time entry restrictions of goods vehicles, we have used the share of goods vehicles (HCV and
LCV) from the study by Errampalli et al. (2020). We use a constant share of fuel type, engine
type and Euro type across all road links. The availability of detailed traffic data, though
challenging, can improve the emission estimates.
Although the COPERT emission functions provide the speed dependent emission factors for
various classes of vehicles, they have been developed for European conditions. This adds to
uncertainties while applying for Indian vehicles. The COPERT speed dependent EFs are
available only for the criteria pollutants such as PM, CO, $NO_x$ and VOC. The emission factors
used here are functions of average speed for each hour. These do not account for the emission
errors due to the speed fluctuations caused due to real-time driving behaviour (frequent
braking, acceleration, deceleration and idling) of the vehicles (Lejri et al., 2018; Lyu et al.,
2021). We have tried to address these by adding another 20% emission across all roads based
on the earlier study (Lejri et al., 2018), however these could be uncertain but are within the
range of uncertainty.
This study only focuses on the hot emissions and does not include cold start, evaporative
emission. We don't consider change in the emissions due to the change in the ambient
temperature and humidity (Franco et al., 2013). Additionally, we don't consider emissions
associated with road slope, vehicle degradation and maintenance in detail. But we have
considered the vehicle degradation effect occurring in older vehicles considering the mileage
as discussed in the COPERT-5 guidebook.
Non-exhaust particulate matter emissions, such as dust resuspension, BW (Brake wear), TW
(Tire wear), RW (Road wear) have not been considered in this study because of larger
uncertainty. However, the non-exhaust emission of PM will be the dominant source of PM
pollution in Delhi (Sharma et al., 2016; TERI, 2018; Singh et al., 2020).
Residential roads, the small roads in residential areas, account for 80% of the total length of
Delhi, however their emission share has been reported to be only ~3% (Singh et al., 2018). We
did not use these roads in our study, firstly, because of small share, secondly, we did not have
a good quality data and thirdly, we wanted to optimise the computational cost.
We reported annual average emissions by considering weekdays and weekends traffic
variations (Figure 2). We did not consider monthly variations as they are much smaller than
the hourly variations. For example, CoV of the EDGAR (Emissions Database for Global
Atmospheric Research; Crippa et al., 2020) monthly emission data over Delhi (shown in Figure
S4) is around 2.5-3% for CO, NMVOC (Non-Methane Volatile Organic Carbon), $NO_x$ and
$PM_{2.5}$ whereas we estimate hourly CoV of 54%, 55%, 19% and 26% for CO, VOC, $NO_x$ and
PM respectively. We do consider the weekdays and weekends traffic variation as they have
substantial variations (Figure 2). Moreover, the hourly weekend and weekdays congestion from
TOMTOM was available as annual mean for 2018, therefore we estimated the annual average
hourly emissions which was converted into annual emissions by summing the hourly emissions
to get daily emissions and then multiplying with 365.
The emissions estimated in this study for Delhi are comparable to the emission estimated for
other megacities. For e.g. road transport emission of $NO_x$ and PM2.5 for London was 20.8 Gg
and 1.12 Gg respectively in 2016 (LAEI, 2016). The megacity Beijing, which has three times
larger road network, had 4.1 Gg of traffic PM emission in 2013 (Jing et al., 2016). While our
estimates are comparable to other megacities, these are lower as compared to the one reported
by earlier studies for Delhi (Table 6). The lower emissions for Delhi can be expected because
India has implemented the recent emission standards in a phased manner (Table S3) which
should reflect in the traffic emission calculations. In many parts of the world, the road transport
emission has decreased, despite an increase in transport vehicles, because of the improvements
in engine technology (Winkler et al., 2018, Sun et al., 2019). One of the reasons for higher
emission estimation by earlier studies for Delhi is the use of old EFs developed by ARAI way
back in 2008. Therefore, these ARAI EFs tend to overestimate the emissions as it does not
represent the recent emission standard technologies (i.e. Euro 3 and Euro 4). It is important to
use recent emission factors such as COPERT-5 which can account for technology related
emissions. Although we have considered advanced traffic flow data and estimated the hourly
emission as a function of speed, the accuracy of the emissions is subject to quality of the input
data and emission factors. Supplying a quality input data and removing ambiguity can improve
the emission estimates and reduce the input data related uncertainty.
Table 6. Traffic emission studies over Delhi.

| Studies | Area | Year | Method | EF | Diurnal | Resolution | PM (Gg) | BC (Gg) | OM (Gg) | CO (Gg) | NO$_x$ (Gg) | VOC (Gg) | NH$_3$ (Gg) | N$_2$O (Gg) | CH$_4$ (Gg) |
|---|---|---|---|---|---|---|---|---|---|---|---|---|---|---|---|
| Das and Parikh (2004) | Delhi | 2005 | VKT | ARAI | NO | - | 5.4 | | | 203 | 39 | | | | |
| Nagpure et al. (2012) | Delhi | 2005 | VKT | Variety of emission factor | NO | - | 10 | | | 350 | 104 | 221 | | | |
| Goyal et al. (2012) | Delhi | 2008 | VKT | IVE | Yes | 2 km | 5.3 | | | 186 | 71 | | | | |
| CPCB (2010) | Delhi | 2010 | VKT | ARAI | NO | 2 km | 3.5 | | | | 30.73 | | | | |
| Sahu et al. (2010, 2015) | NCR Delhi | 2010 | VKT | ARAI | NO | 1.67 km | 30.3 | | | 427 | 162 | | | | |
| Guttikunda and Calori (2013) | NCT Delhi | 2010 | VKT | ARAI and Other | NO | 1 km | 14 | | | 256 | 199 | 132 | | | |
| Singh et al. (2018) | NCT Delhi | 2010 | Non-VKT | ARAI | NO | 100 m | 4.5 | | | 114 | 51.5 | | | | |
| Goel et al. (2015a) | NCT Delhi | 2012 | VKT | COPERT-3 and ARAI | NO | - | 12.7 | | | 300 | 184 | 71.6 | | | |
| Sharma et al. (2016) | NCT Delhi | 2014 | Non-VKT | ARAI | NO | 2 km | 4.7 | | | 117 | 41.5 | | | | |
| TERI (2018) | NCT Delhi | 2016 | | ARAI | NO | 4 km | 12.4 | | | 501 | 126 | 342 | | | |
| SAFAR (2018) | NCR Delhi | 2018 | VKT | ARAI | NO | 400 m | 43.2 | 15.5 | | 483.1 | 257.7 | 614.5 | | | |
| This Study | NCT Delhi | 2018 | Non-VKT | COPERT-5 | YES | 100 m | 1.82 | 0.94 | 0.75 | 221 | 56 | 64 | 0.28 | 0.26 | 11.38 |

* NCT area is around 1483 km$^2$; NCR area is around 4550 km$^2$.

**6 Conclusion**

Here we present a methodology to estimate high-resolution spatially resolved hourly traffic
emission over Delhi using advanced traffic flow and speed. We estimated the emissions of
major pollutants, viz. PM, BC, OM, CO, $NO_x$, VOC, $NH_3$, $N_2O$ and $CH_4$.
We have used traffic volume and speed measurements conducted at 72 locations over Delhi in
the year 2018 as a part of TRIPP of IIT Delhi. Additionally, we have used the hourly congestion
data from TomTom to account for hourly changes in the speed. The studies relation between
traffic volume and speed has been utilised to generate the hourly traffic volume and speed
profile for each road link. The vehicles have been classified into 127 categories according to
vehicle types, fuel type, engine capacity, emission standard. The COPERT-5 emission
functions of speed are applied at a micro level for each hour along each road link to calculate
the emissions that accounts for congestion and spatial variation in emission. To the best of our
knowledge, this is the first study of its kind which considers advanced traffic flow data and
estimates the hourly multi-pollutant emissions as a function of speed. We make the following
conclusions:
1. We estimated an annual emission of 1.82 Gg for PM, 0.94 Gg for BC, 0.75 Gg for OM,
221 Gg for CO, 56 Gg for $NO_x$, 64 Gg for VOC, 0.28 Gg for $NH_3$, 0.26 Gg for $N_2O$ and
11.38 Gg for $CH_4$ in 2018. We estimated an uncertainty of 60%- 68% in these emissions
by adding 40% uncertainty in VKT and 80% uncertainty in EFs.
2. The modelled traffic volume (in PCU) and speed profiles show bimodal distribution
exhibiting an anti-correlation behaviour. The traffic volume peaks during morning and
evening rush hours resulting in lower speed. There is a mild enhancement in speed during
the afternoon due to the less traffic. During the early morning hours, the vehicles almost
achieve the free flow speed.
3. The diurnal variation of emission of pollutants are like traffic variations and show distinct
bimodal distribution with morning and dominant evening peaks for almost all pollutants.
However, the difference in night-time and day-time emissions are less for PM, BC and $NO_x$
due to the enhanced share of goods vehicles during the night-time. The good vehicles
significantly contribute to the night-time emission in Delhi. These emissions along with
unfavourable meteorology (e.g. lower PBL and wind speed) might help in sustained PM
levels during the night-time in Delhi.
4. In terms of the spatial distribution of the emissions, the emissions are higher along the
major roads and the emission hotspots are near the traffic junctions. The emission flux in
inner Delhi is highest due the higher road and traffic density, and lower average speed. This
is 40-50% higher than the mean emission flux of Delhi. However, the total emission is
higher for outer Delhi due to its larger area having a total road length more than inner Delhi.
5. According to the road classes (RClass1 to RClass5, from single lane to multi-lane roads),
we find that RClass3 has the highest emission share due to highest total road length.
However, the emission per km is highest over multi-lane wider roads (RClass4 and
RClass5) that is almost two times RClass3 because of high traffic volume. Moreover, the
emission per lane per kilometre is highest for RClass1 because of lower speed and
congestion. While the effective management of traffic in narrow roads could be beneficial,
the multi-lane roads act as emission hotpots. An analysis of the choice of road width should
be performed to achieve the optimum emission without increasing the pollution exposure
near the roads.
6. Petrol vehicles contribute to over 50% emission of OM, CO, VOC, $NH_3$, $N_2O$ and $CH_4$
emissions. For OM, CO, VOC, $N_2O$ and $CH_4$ the petrol share is dominated by 2W whereas
for $NH_3$, share is dominated by petrol cars. The diesel vehicles are the dominant contributor
to PM, BC and $NO_x$ emission.
7. In terms of emission standards, Euro3 vehicles contribute the highest to all pollutants
followed by Euro4 with an exception to $NH_3$ where Euro2, mainly petrol cars, are the
dominant source.
8. Among vehicle classes, the 2Ws contribute the most to the total emissions for all the
pollutants except for BC, $NO_x$ and $N_2O$. The diesel vehicles including goods vehicles (HCV
and LCV) contribute substantially to the PM, BC and $NO_x$ emissions. The goods vehicles
have a dominant share in the night-time emissions. CNG Buses have the highest
contribution to $NO_x$ and $CH_4$ emissions whereas diesel Buses have substantial contributions
to PM emissions. Petrol cars are the dominant source for $NH_3$ whereas diesel cars contribute
substantially to PM, BC and $NO_x$ emissions. The contribution of petrol cars to the PM
emission is less than 2%.
9. For all the pollutants, the top 5 polluting vehicle categories account for more than half (55%
- 91%) of the emissions. The pollutants such as CO, VOC, $CH_4$ and OM have a distinct
source such as 2W. However, the PM and BC have mixed sources including 2W and diesel
vehicles. $NO_x$ emissions are mainly due to CNG and diesel vehicles. $NH_3$ is mainly emitted
from petrol and diesel cars and $N_2O$ has mixed sources including 2W and cars.
This spatio-temporal emissions can be used in air quality models for developing suitable
strategies to reduce the traffic related pollution in Megacity Delhi. Moreover, the developed
methodology is a step forward in developing real-time emission prediction in the future with
growing availability of real-time traffic data.
**Data availability**
The emission dataset can be accessed through the open-access data repository
https://doi.org/10.5281/zenodo.6553770 (Singh et al., 2022), under a CC BY-NC-ND 4.0
license. This dataset is presented as a netCDF covering the rectangular domain around National
Capital Territory (NCT) of Delhi. The data and analysis presented in the paper is only over the
NCT area as shown in Figure 3. TOMTOM averaged congestion data is available online
(https://www.tomtom.com/en_gb/traffic-index/new-delhi-traffic/). COPERT-5 emission
factors are obtained from the EMISIA online platform
(https://www.emisia.com/utilities/copert/) of Aristotle University, Thessaloniki.
**Author contribution**
**Vikas Singh** and **Akash Biswal**: Conceptualization, investigation, visualization, formal
analysis, writing original draft, writing, reviewing and editing; **Leeza Malik** and **Geetam**
**Tiwari**: Traffic data validation, investigation, discussion, reviewing and editing; **Ravindra**
**Khaiwal** and **Suman Mor**: Investigation, discussion, reviewing and editing.
**Declaration of competing interest**
The authors declare that they have no conflict of interest.
**Acknowledgments**
The authors are thankful to the Director, National Atmospheric Research Laboratory (NARL,
India), for encouragement to conduct this research and provide the necessary support. AB is
thankful to the Department of Environment Studies, Panjab University, Chandigarh for
providing the necessary support and greatly acknowledges the MoES (Ministry of Earth
Sciences, India) for providing support as a part of PROMOTE project. Authors greatly
acknowledge the Transportation Research and Injury Prevention Programme (TRIPP) of IIT
Delhi to provide the advanced traffic data. We acknowledge and thank TOMTOM for making
available the congestion profile over Delhi. We acknowledge the EMISIA platform of the
Aristotle University of Thessaloniki for providing the COPERT-5 emission factor.  This paper

is based on interpretation of results and in no way reflects the viewpoint of the funding agencies.

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
