# Peer review of "Spatially resolved hourly traffic emission over megacity Delhi"

_Earth System Science Data, 2022_

## Author Comment (AC1)

**Authors' responses to Referee #1.**

Reviewer's comments are in black text and author's responses are in blue text.

**Anonymous Referee #1**

This manuscript describes a high-resolution road traffic emission inventory developed for the megacity of Delhi using advanced and detailed traffic data and speed based EFs. The strength of the estimation methodology presented in the paper is in its usage of very detailed and advanced input datasets, which allows obtaining high-resolution spatio-temporal emission maps and disaggregate the emission results according to several categories, including vehicle types, road classes or hours of the day. The resulting dataset is therefore relevant for policy makers, but also for air quality who want to use it as input in the chemical transport models. The paper is well written and structured, and its quality is very good, which makes it a good contribution to ESSD. However, there are some aspects related to the methodology proposed that should be better clarified before the manuscript is accepted for publication.

We thank the referee #1 for taking time to review the manuscript. We appreciate the positive feedback and valuable comments that have helped to improve the manuscript.

Particular Comment:

1. Hourly congestion data from TomTom is used to estimate traffic flow information per road link following equation 3, which is presented in section 2.1.1. According to this equation, if congestion is 0, the resulting traffic flow will also be 0. Nevertheless, null congestion does not imply having no cars circulating. Can you clarify how this issue is corrected in the model?

   We agree with the reviewer's comment that null/zero congestion does not imply zero traffic on the road. We thank the reviewer for pointing out this issue in equation 3 which requires further clarification.

   Congestion is defined as the percentage travel time delay, that is the extra time required to complete a trip. In real world situations, even with the light traffic the congestion exists where minimum time delay is observed in order to reduce the likelihood of collision, known as single interaction (Vickrey, 1969). Therefore the congestion can not be zero in large cities such as Delhi with complex urban geometry and nighttime activity. The night time traffic can be considered as a smooth traffic flow situation with a low congestion value. Therefore, in order to avoid zero traffic, we have used a minimum congestion value of 0.03 (3%) for Delhi to match the nighttime traffic levels reported by (Errampalli et al. 2020). Moreover, a similar level of nighttime congestion has been reported by Wei et al. (2022) in the large Chinese cities. We have clarified this in the manuscript Line 200 - 219 section 2.1.1.

2. The resulting traffic flow information is validated by comparing estimated and reported annual VKT information. However, results from this comparison are not provided. Please add them.

The comparison between estimated annual VKT and reported by other studies has now been provided in the text in section 3.1 and tabulated in Table S11 in the supplementary material and Table 1 in response. This table (Table 1) includes the studies which have either reported annual VKT or have provided enough data to calculate annual VKT.

The VKT values compare well with the earlier studies by considering the fact that the uncertainties exist in the method of estimation, year and study domain. Malik et al. (2019) estimated the destined and non-destined VKT of freight vehicles (HCV and LCV) with the actual measured traffic at several entry points in Delhi. Goel et al. (2015b) estimated the annual VKT based on the annual mileage of the 2W and cars obtained from PUC (Pollution under control) certification data and the number of registered vehicles. The VKT reported by Goel et al. (2015b) for Cars and 2W are slightly lower than our study. The study by Goel et al. was conducted in 2012 since then the  cars and taxis share has almost doubled in Delhi due to increased travel demand and economic growth (DDA 2021). The study by Kumar et al. (2011), which is for 2010, reported higher VKT for Buses and HCV as compared to the one estimated by the current study. Their estimates were based on the assumed distance traveled by each vehicle and the number of registered vehicles than the actual on road vehicle. Guttikunda  and  Calori. (2013) reported high VKT for buses and HCV. The study by Sahu et al. (2011) for NCR Delhi estimated very high VKT for 2W and Cars. While earlier studies have reported different VKT values,  the relative VKT share compares well with our study. Moreover, the VKT estimated by recent studies are close to our estimates.

Table 1. Comparison of the VKT (in billion VKT) current study with the previous studies over Delhi.

| Vehicle category | Current Study | Malik et al. (2019) | Goel et al. (2015b) | Kumar et al. (2011) | Guttikunda and  Calori., (2013) | Sahu et al., (2011) |
|---|---|---|---|---|---|---|
| Study year | 2018 | 2016 | 2012 | 2010 | 2010 | 2010 |
| 2W | 31.63 | | 24.73 | 24.05 | | 70.8 |
| 3W | 6.11 | | | 3.68 | 2.42 | 1.8 |
| CAR | 27.36 | | 16.56 | 29.82 | 22.4 | 57.8 |
| Buses | 1.71 | | | 5.01 | 6.7 | 2.8 |
| HCV | 0.95 | 0.99 | | 2.94 | 4.02 | 4.2 |

| | | | | |
|---|---|---|---|---|
| LCV | 3.14 | 4.44 | 3.59 | 4.6 |

Also, could you provide a comparison between estimated and measured hourly traffic flow for those locations in which you have observations? This will give to the reader a better feeling of how robust this approach is.

As suggested, we have provided the comparison between estimated and measured hourly traffic (8 am - 2 pm ) at 72 locations (Fig. 1 of this response, also shown in Fig. S3). The estimated and measured traffic have a correlation of 0.99 and the difference (estimated - measured) varies from -0.6% to 2.6%.

[Figure]

Figure 1. Percentage difference (100%*[estimated - measured]/measured) between estimated and observed traffic in terms of PCU at the 72 locations.

3. According to the authors, "the emissions are further adjusted with a factor of 1.2 to account for real-time driving behaviour (frequent braking, acceleration, deceleration) as per the study by Lejri et al., (2018)". This assumption seems to me a bit arbitrary a not well justified. Is this factor applied to all hours of the day (including those when congestion is low)? Is this factor applied equally to all pollutants? Why?

We agree with the reviewer that the real-world emissions are highly uncertain due to spontaneous speed fluctuations caused due to real-time driving behaviour (frequent braking, acceleration, deceleration. These depend on emitted pollutants, vehicle type, fuel type, driving conditions etc. COPERT relies on mean driving speed and travel distance. The mean speeds are relatively low under urban driving conditions, and emission factors are highly variable within this speed range due to the speed fluctuations.

Because of its complexity, different authors have reported different correction factors after comparing it with real world emissions. The study conducted for Indian cities by Mahesh et al. (2018) reported significant increase in emission rate with acceleration for all the test cars. According to a study conducted by Bokare and Maurya (2013) on the effect of acceleration and deceleration on passenger car emissions on Indian highways, acceleration of 2 $m/s^2$ can increase emissions by double for HC and $NO_x$. A study on passenger automobiles in Delhi by Jaiprakash and Habib (2017) found that the emission rate varies for different fueled vehicles and can be up to ten times higher ($NO_x$ and CO) during the acceleration/deceleration range of -1 to 1 $m/s^2$. Davison et al. (2021) measured emissions under real driving conditions to develop new bottom-up inventories and compared to official national inventory totals. They found that the total UK passenger car and light-duty van emissions of nitrogen oxides ($NO_x$ ) are underestimated by 24–32%. Lejri et al. (2018) has studied the impact of variations in the estimated mean speeds on the emission factors estimated within COPERT. They have estimated the relative errors on fuel consumption and $NO_x$ emissions related to mean speed variations from 2 to 10 km/h and estimated errors up to 25-30% in fuel consumption and $NO_x$ emissions. Samaras et al (2019) estimated fuel consumption from vehicles circulated on urban roads with different levels of congestion with an aim to refine the average speed model (COPERT) functions and showed that under congested conditions the fuel consumption can increase by more than 18%.

Therefore to account for the emissions due to the speed fluctuations around the mean speed, a factor of 1.2x, i.e. 20% increase has been applied to the final dataset. This has been applied for all the hours and all the pollutants. Fig. 2 of this response shows the hourly variation of extra emission that we have added. Although we apply the same factor for all hours of the day, the added emissions are more during high congestion hours and less during low congestion hours. The total added emission is also different for different pollutants.

We agree that this factor is uncertain and due to the lack of a suitable correction factor for Indian conditions, we have chosen a fixed factor to increase the emissions. The readers or the emission data user may be able to remove this dividing by 1.2 and use their own correction factor in future studies. This is also one of the limitations of the study which has been discussed in the limitation Section 5 of the manuscript.

[Figure]

Figure 2. Additional emissions (20%) in Mg/hour to account for the speed fluctuation around the mean speed

4. As mentioned by the authors, emissions are estimated using COPERT 5, which is a European emission model and has not been calibrated for Indian conditions. Can the authors elaborate a bit more on the potential uncertainty for key vehicle categories such as two-wheeler motor bikes? Perhaps the EFs reported by COPERT could be compared against results reported by local studies such as Adak et al., 2016, https://doi.org/10.1016/j.scitotenv.2015.11.099.

We agree with the reviewer's comment that COPERT being a European model, is not calibrated to Indian vehicles. However, it is to be noted that Indian emission norms are in line with European emission norms (ARAI 2008; https://morth.nic.in/vehicular-emission-norms; Singh et al., 2022). COPERT emission factors are functions of speed and have potential advantages as compared to static emission factors of ARAI (2008), as it can capture the emission change at varying speeds during a day.

In order to elaborate upon the potential uncertainty in the key vehicle categories, we have compared the COPERT EFs used in this study with the earlier reported EFs and shown in Table

2 in response and Table S12 in supplementary material. In case of 2W measured EF of CO, HC and $NO_x$ has a range of 1 to 6.7, 0.33 to 0.45 and 0.21 to 0.46 g/km respectively, which are within the range of COPERT EFs. Similarly for the passenger cars the COPERT EF has a good agreement with the values reported by Jaiprakash et al., (2018) and Jaikumar et al., (2017). The CO emission factor reported by Adak et al. (2016) is very low compared to all measured studies and the COPERT EF. The CoV of EF reported in Table S12 varies from 40% to 120%, therefore we consider an uncertainty of ~80% in the EFs across all pollutants and vehicles.

Further, we have made an attempt to estimate the uncertainty in emissions by introducing uncertainty in VKT and EF. Based on the reported VKT and EF by earlier studies as shown in Table 2 and Table 1 respectively, we estimated an uncertainty of ~40% and ~80% in VKT and EF respectively. Then we calculated the total emission of pollutants by varying the VKT from -40% to +40% of the VKT used and by varying the EF from -80% to +80% with an interval of 10%. The obtained distribution of the emission of pollutants is shown in Fig. 3 of this response below and Fig. 7 in the main manuscript. We calculated the CoV (Coefficient of Variation, CoV = [Std/Mean]*100%) of the distribution and estimated an uncertainty of 61%, 60%, 63% and 68% for CO, PM, $NO_x$ and VOC respectively. Dey et al., (2019) had estimated uncertainties of the emission of CO, VOC and NMVOC for Ireland in the range of −58% to +76%. Kouridis et al. (2010) estimated coefficient of variation of 10% for $CO_2$, in the order of 20-30% for $NO_x$, VOC, $PM_{2.5}$, $PM_{10}$, 50-60% for CO and $CH_4$ and over 100% for $N_2O$.

Now we have included a separate section (4) *Uncertainty in emission* in the manuscript to explain the emission uncertainty with input parameters.

Table 2. COPERT emission factor comparison with the measured emission factor in Indian condition

| Vehicle Type | | 2W (g/km) | | | | Diesel Car (g/km) | | | | Petrol Car (g/km) | | | |
|---|---|---|---|---|---|---|---|---|---|---|---|---|---|
| Studies | Location | CO | HC | NOx | PM | CO | HC | NOx | PM | CO | HC | NOx | PM |
| This Study COPERT at 34 km/h (min to max) | COPERT applied for Delhi | 2.2 (0.15 - 12.8) | 0.49 (0.037 - 3.74) | 0.07 (0.017 - 0.33) | 0.01 (0.005 - 0.04) | 0.21 (0.044 - 0.73) | 0.035 (0.007 - 0.26) | 0.69 (0.43 - 1.33) | 0.037 (0.025 - 0.073) | 0.46 (0.147 - 1.853) | 0.036 (0.01 - 0.25) | 0.10 (0.037 - 0.26) | 0.002 (0.0013 - 0.0032) |
| TERI/ARAI 2018 | Indian Vehicle | 0.39 - 4.3 | 0.25 - 1.18 | 0.16 - 0.30 | 0.002 - 0.04 | 0.15 - 0.38 | 0.03 - 0.07 | 0.29 - 0.54 | 0.008 - 0.03 | 0.82 - 0.98 | 0.07 - 0.09 | 0.04 - 0.07 | 0.001 - 0.002 |
| Kuppili et al., 2021 | Delhi | | | | | 3.99 | 0.34 | 0.54 | | 7.26 | 0.17 | 0.62 | |
| Jaiprakash et al., 2018 | Delhi | 1.0 ± 0.6 | | 0.07 ± 0.01 | | 0.3 ± 0.1 | | 1 ± 0.4 | | 2.2 | | 1 | |
| Jaiprakash et al., 2017 | Delhi | | | | 0.014 | | | | 0.19 | | | | 0.067 |
| Adak et al., 2016 | Dhanbad | 1.6 | 0.45 | 0.46 | | 0.07 | 0.1 | 0.35 | | | | | |
| Mahesh et al., 2018 | Chennai | | | | | 1.28 | 0.13 | 0.59 | | | | | |
| Mahesh et al., 2019 | Chennai | 6.7 ± 2.60 | 0.33 ± 0.12 | 0.217 ± 0.032 | | | | | | | | | |
| Jaikumar et al 2017 | Chennai | | | | | 0.6 | 0.06 | 1.3 | | | | | |

[Figure]

Figure 3. Histogram shows the variation of emission with the combination of sensitive parameters

5. The vehicular classification is done making use of shares provided by different local sources of information. Are these shares based on information of registered vehicles or actual circulating vehicles? (i.e., old vehicles may appear as registered by they are barely used in reality)

The primary vehicle classification such as 2W, 3W, cars, buses, LCV and HCV for each roadlink is based on the TRIPP measured traffic data. For further sub classes such as fuel type, engine type is based on published literature and reports. For the Euro classification, we have used the vehicular survival function (Goel et al., 2015b; Malik et al., 2019) and calculated the Euro share based on the Euro implementation year and the number of registered vehicles. The vehicle survival was calculated for the past twenty years by considering 2018 as the base year and then the Euro share was calculated based on the age of the vehicle with respect to 2018. The same has been described in *section 2.2.*

We consider the actual circulating vehicles based on the TRIPP survey data. We use the survival function to retain the share of the new vehicles as the old vehicles have reached the end of their life. The same can be evident from Table S4 where 84% of the cars in 2018 are Euro 4. For other vehicles, Euro 4 was implemented in 2017-2018 (Table S3), therefore more than 80% of vehicles are either Euro 3 or Euro 4.

6. According to the authors, "annual emissions have been calculated by summing the hourly emissions to get daily emissions and then multiplying with 365". By doing this, authors are assuming that for all days of the week (Monday to Sunday) and all months of the year (January to December) traffic activity and emissions present the same intensity. However, traffic activity and associated emissions typically present a drop during weekends when compared to weekdays, and they can also present drops/increases during certain months of the year. Is this not the case for Delhi? Can the authors provide some information that support they hypothesis (i.e. emissions are constant throughout the year).

We agree with the reviewer's comment that traffic emission is not consistent throughout the year. Road traffic emission is highly dependent on the traffic flow that has temporal patterns which can be monthly, weekdays and weekend, and hourly. However, the monthly variations are much smaller than the hourly variations. For example, coefficient of variation (CoV = [Std/ Mean]*100% ) of the EDGAR (Emissions Database for Global Atmospheric Research; Crippa et al., 2020) monthly emission data over Delhi (shown below in Fig. 3 anda also in Fig. S4 in supplementary material) is around 2.5-3% for CO (Carbon Monoxide), NMVOC (Non Methane Volatile Organic Carbon), $NO_x$ (Oxides of Nitrogen) and $PM_{2.5}$ whereas we estimate hourly CoV of 54%, 55%, 19% and 26% for CO, VOC, $NO_x$ and PM respectively (Table 3). We do consider the weekdays and weekends traffic variation as they have substantial variations (as shown Fig. 2 of the main manuscript). Moreover the hourly weekend and weekdays congestion from TOMTOM was available as annual mean for 2018, therefore we estimated the annual average hourly emissions which was converted into annual emissions by summing the hourly emissions to get daily emissions and then multiplying with 365. We will be willing to calculate the monthly emission in our future studies when we have more data available. This has been added in the limitation in section 5.

[Figure]

Figure 3. Monthly emission variation around Delhi for CO, NMVOC, $NO_x$ and $PM_{2.5}$.

Table 3. Statistics of monthly and hourly emission

| Monthly variation | | Hourly variation | |
|---|---|---|---|
| Pollutant | CoV (%) | Pollutant | CoV (%) |
| CO | 2.8% | CO | 54.2% |
| NMVOC | 2.5% | VOC | 55.6% |
| $NO_x$ | 2.8% | $NO_x$ | 19.3% |
| $PM_{2.5}$ | 2.8% | PM | 25.9% |

*CoV: Coefficient of Variation (100\*std/mean)*

**Other comments:**

1. Particulate matter emissions are usually expressed as PM (regardless of the fact that they include or not non-exhaust emissions). I would recommend to change the acronym from PME to PM - In the text the authors already specify that PM emissions only include exhaust - and also specify if this PM equals PM10 and/or $PM_{2.5}$.

    We agree, we have modified the acronym from PME to PM throughout the manuscript. The exhaust PM is mostly less than 2.5 µm (Pant and Harrison 2013) and around 98% of them are $PM_{2.5}$ (ARAI 2008). The same has been updated in the manuscript Line 142.

2. "In this study we have shown a data driven approach where the quality of input data is likely to improve the emission estimates." I believe this is a too strong conclusion. Emission results from this work differs significantly from previous estimates as it makes use of more updated and refined information, but it cannot be concluded that the estimates have been improved. In order to say that, an evaluation of the emission dataset should be performed by, for instance, performing an air quality modeling study and comparing the results against observations.

    We agree with the reviewer's comment that results from this work differs significantly from previous estimates as it makes use of more updated and refined information, however we also understand the uncertainties involved in such detailed emission calculations. Therefore, we stressed that the emission estimation is a data driven approach and there is a scope to further improve the emission estimates by providing detailed quality data as an input to the emission model. Hence providing detailed quality input data is likely to improve the emission estimates. In order to claim that we have improved the emission, we agree with the reviewer that an evaluation and intercomparison of the available emissions needs to be performed which will be taken as future studies. Moreover, the developed methodology is a step forward in developing real time emission with the growing availability of real-time traffic data.

**Reference**

Adak, P., Sahu, R., and Elumalai, S. P.: Development of emission factors for motorcycles and shared auto-rickshaws using real-world driving cycle for a typical Indian city, Science of The Total Environment, 544, 299–308, https://doi.org/10.1016/j.scitotenv.2015.11.099, 2016.

ARAI.: Automotive Research Association of India, Development of emission factor for Indian vehicles in the year 2008, Air Quality Monitoring Project-Indian Clean Air Programme (ICAP), pp. 1-89, http://www.cpcb.nic.in/Emission_Factors_Vehicles.pdf, 2008.

Bokare Shridhar, P. and Maurya Kumar, A.: STUDY OF EFFECT OF SPEED, ACCELERATION AND DECELERATION OF SMALL PETROL CAR ON ITS TAIL PIPE EMISSION, IJTTE, 3, 465–478, https://doi.org/10.7708/ijtte.2013.3(4).09, 2013.

Crippa, M., Solazzo, E., Huang, G., Guizzardi, D., Koffi, E.,Muntean, M., Schieberle, C., Friedrich, R., and Janssens-Maenhout, G.: High resolution temporal profiles in the EmissionsDatabase for Global Atmospheric Research, Sci. Data., 7, 121,https://doi.org/10.1038/s41597-020-0462-2, 2020.

Davison, J., Rose, R. A., Farren, N. J., Wagner, R. L., Murrells, T. P., and Carslaw, D. C.: Verification of a National Emission Inventory and Influence of On-road Vehicle Manufacturer-Level Emissions, Environ. Sci. Technol., 55, 4452–4461, https://doi.org/10.1021/acs.est.0c08363, 2021.

DDA: Baseline report for transport: Delhi Development Authority and National Institute of Urban Affair, Master Plan for Delhi 2041, https://online.dda.org.in/mpd2041dda/_layouts/MPD2041FINALSUGGESTION/Baseline_Transport_%20160721.pdf, 2021.

Dey, S., Caulfield, B., and Ghosh, B.: Modelling uncertainty of vehicular emissions inventory: A case study of Ireland, Journal of Cleaner Production, 213, 1115–1126, https://doi.org/10.1016/j.jclepro.2018.12.125, 2019.

Goel, R., Guttikunda, S. K., Mohan, D., and Tiwari, G.: Benchmarking vehicle and passenger travel characteristics in Delhi for on-road emissions analysis, Travel Behaviour and Society, 2, 88–101, https://doi.org/10.1016/j.tbs.2014.10.001, 2015.

Guttikunda, S. K. and Calori, G.: A GIS based emissions inventory at 1 km × 1 km spatial resolution for air pollution analysis in Delhi, India, Atmospheric Environment, 67, 101–111, https://doi.org/10.1016/j.atmosenv.2012.10.040, 2013.

Jaikumar, R., Shiva Nagendra, S. M., and Sivanandan, R.: Modeling of real time exhaust emissions of passenger cars under heterogeneous traffic conditions, Atmospheric Pollution Research, 8, 80–88, https://doi.org/10.1016/j.apr.2016.07.011, 2017.

Jaiprakash and Habib, G.: Chemical and optical properties of PM2.5 from on-road operation of light duty vehicles in Delhi city, Science of The Total Environment, 586, 900–916, https://doi.org/10.1016/j.scitotenv.2017.02.070, 2017.

Jaiprakash and Habib, G.: On-road assessment of light duty vehicles in Delhi city: Emission factors of CO, CO2 and NOX, Atmospheric Environment, 174, 132–139, https://doi.org/10.1016/j.atmosenv.2017.11.039, 2018.

Kouridis, C., Gkatzoflias, D., Kioutsioukis, I., Ntziachristos, L., Pastorello, C. and Dilara, P.: Uncertainty estimates and guidance for road transport emission calculations: Publications Office, LU, https://publications.jrc.ec.europa.eu/repository/bitstream/JRC57352/uncertainty%20eur%20report%20final%20for%20print.pdf, 2010.

Kousoulidou, M., Fontaras, G., Ntziachristos, L., Bonnel, P., Samaras, Z., and Dilara, P.: Use of portable emissions measurement system (PEMS) for the development and validation of passenger car emission factors, Atmospheric Environment, 64, 329–338, https://doi.org/10.1016/j.atmosenv.2012.09.062, 2013.

Kumar, P., Gurjar, B. R., Nagpure, A. S., and Harrison, R. M.: Preliminary Estimates of Nanoparticle Number Emissions from Road Vehicles in Megacity Delhi and Associated Health Impacts, Environ. Sci. Technol., 45, 5514–5521, https://doi.org/10.1021/es2003183, 2011.

Kuppili, S. K., Dheeraj Alshetty, V., Diya, M., Shiva Nagendra, S. M., Ramadurai, G., Ramesh, A., Gulia, S., Namdeo, A., Maji, K., Bell, M., Goodman, P., Hayes, E., Barnes, J., Longhurst, J., and De Vito, L.: Characteristics of real-world gaseous exhaust emissions from cars in heterogeneous traffic conditions, Transportation Research Part D: Transport and Environment, 95, 102855, https://doi.org/10.1016/j.trd.2021.102855, 2021.

Mahesh, S., Ramadurai, G., and Shiva Nagendra, S. M.: Real-world emissions of gaseous pollutants from diesel passenger cars using portable emission measurement systems, Sustainable Cities and Society, 41, 104–113, https://doi.org/10.1016/j.scs.2018.05.025, 2018.

Mahesh, S., Ramadurai, G., and Shiva Nagendra, S. M.: Real-world emissions of gaseous pollutants from motorcycles on Indian urban arterials, Transportation Research Part D: Transport and Environment, 76, 72–84, https://doi.org/10.1016/j.trd.2019.09.010, 2019.

Malik, L., Tiwari, G., Thakur, S., and Kumar, A.: Assessment of freight vehicle characteristics and impact of future policy interventions on their emissions in Delhi, Transportation Research Part D: Transport and Environment, 67, 610–627, https://doi.org/10.1016/j.trd.2019.01.007, 2019.

Pant, P. and Harrison, R. M.: Estimation of the contribution of road traffic emissions to particulate matter concentrations from field measurements: A review, Atmospheric Environment, 77, 78–97, https://doi.org/10.1016/j.atmosenv.2013.04.028, 2013.

Sahu, S. K., Beig, G., and Parkhi, N. S.: Emissions inventory of anthropogenic PM2.5 and PM10 in Delhi during Commonwealth Games 2010, Atmospheric Environment, 45, 6180–6190, https://doi.org/10.1016/j.atmosenv.2011.08.014, 2011.

Samaras, C., Tsokolis, D., Toffolo, S., Magra, G., Ntziachristos, L., and Samaras, Z.: Enhancing average speed emission models to account for congestion impacts in traffic network link-based simulations, Transportation Research Part D: Transport and Environment, 75, 197–210, https://doi.org/10.1016/j.trd.2019.08.029, 2019.

Singh, S., Kulshrestha, M. J., Rani, N., Kumar, K., Sharma, C., and Aswal, D. K.: An Overview of Vehicular Emission Standards, MAPAN, https://doi.org/10.1007/s12647-022-00555-4, 2022.

Vickrey, W. S.: Congestion Theory and Transport Investment, The American Economic Review, 59, 251–260, https://www.jstor.org/stable/1823678, 1969.

Wei, X., Ren, Y., Shen, L., and Shu, T.: Exploring the spatiotemporal pattern of traffic congestion performance of large cities in China: A real-time data based investigation, Environmental Impact Assessment Review, 95, 106808, https://doi.org/10.1016/j.eiar.2022.106808, 2022.

---

## Author Comment (AC2)

**Authors' responses to Referee #2.**

Reviewer's comments are in black text and authors' responses are in blue text.

**Anonymous Referee #2**

This study presents a multi-pollutant hourly gridded vehicle emission inventory over Delhi, with a bottom-up methodology. Hourly congestion data from TomTom were used to account for hourly changes in the speed. The traffic flow was derived from speed based on fitting formula established by the previous research. The percentage share of vehicle technical parameters (vehicle type, fuel used, emission standards, etc.) was provided by survey reports or previous study in Delhi. Emission factors were calculated by COPRET-5. This paper is well-written and presents results that would be interesting to the air quality modeling community or policy makers. However, I have several concerns that the authors should consider when revising the manuscript, as listed below. I recommend this work to be published after the following comments are adequately addressed.

We thank the referee #2 for taking time to review the manuscript. We appreciate the positive feedback and valuable concerns/comments that have helped to improve the manuscript.

**Particular Comments:**

1. Section 2.1.1., Values in Table S2 were different from those given in Malik et al., 2021. This would lead to huge deviations in subsequent calculations. What are the reasons for this revision?

The road links used in this study are classified into five road classes (RClass1 to RClass5) based on the width of the road as per TRIPP report (Malik et al., 2018). Speed–volume relationship for different road classes in Delhi reported by Malik et al. (2021) are given for different lanes (1 lane, 2 lanes, 3 lanes and >4 lanes). In order to harmonize the road classes, we use RClass1 for 1 lane, RClass2 for 2 lanes, RClass3 for 3 lanes, and RClass 4 and RClass 5 for >4 lanes. We selected the parameters of the road classes that have high numbers of sample points and higher $R^2$ corresponding to each road class. For eg, for RClass3, we considered the 3 lanes having higher $R^2$. The values of the corresponding parameters are listed in Table S2. We do, however, acknowledge that the table contained one typo that has now been corrected. The above explanation is now added in the manuscript section 2.1.1 (226 -231).

2. Line 218-220, How to correct the speed and traffic volume, all roads or some specific roads?

The hourly traffic volume and speed for each road link was estimated using the methodology described in section 2.1.1. The hourly congested speed has been calculated using equation 2 and the hourly traffic has been calculated using equation 3 for different types of road classes. This results in activity data as per road classes. As we have TRIPP traffic data (8am to 2pm) for each road link, we corrected the traffic and speed data for each road link by taking the ratio

of estimated and TRIPP traffic data during 8 am to 2 pm to match the observed traffic keeping the hourly variation intact at the time of bias correction. The difference between the observed and corrected estimated hourly traffic (8 am - 2 pm ) at 72 locations is shown in Fig. S3. The estimated and measured traffic have a correlation of 0.99 and the difference (estimated - measured) varies from -0.6% to 2.6%.

3.  Whether the vehicular share (%) was constant for the specific road throughout the day? It was not reasonable. And, this directly determined the result of vehicular volume and emission share (section3.1 and section 3.6).

The vehicle shares of cars, buses, 2W, 3W, LCV and HCV is varying for each hour as well as each road of Delhi. Fig. 1 (b) below (also shown in Figure S1) shows the estimated hourly mean vehicle share over Delhi which is changing. The passenger vehicles namely 2W, 3W and cars have a large share during activity hours (08:00 to 20:00) whereas the night-time traffic is dominated by commercial vehicles due to the restriction of movement of freight vehicles during the peak traffic hours and ban of HCVs during daytime leading to increases share during night hours (23:00 to 5:00).

[Figure]

Figure 1. The stacked percentage bar plot showing the estimated hourly mean traffic composition over Delhi.

4.  Table S4, all 3W vehicles were Euro 4 ?

We have performed this study for the year 2018. As per the official data, 99.8% (~100%) of the 3W (other than electric) were Euro 4. Therefore, we considered all 3W as Euro 4 using CNG fuel as CNG is mandatory in Delhi (Hakkim et al., 2022) so all the 3W are CNG (Sahu et al., 2011; Dhyani and Sharma 2017).

5.  Section 3.3 and 3.4, The authors seem to assume that as long as the road types are the same, the relationship between speed and vehicle flow are the same too. The resulting

spatial distribution may have large errors. Authors should consider making some corrections.

Yes, we use the same speed-volume relationship for the same type of the road, however we further correct the traffic and speed for each road link based on the TRIPP data. This means although the traffic or speed variation for the same type of road is similar, they differ in terms of the traffic composition, count and average speed. We have shown the traffic and speed variations across different road classes as a box plot, shown in Fig. 2 of response (Fig. S5 in supplementary material). While the speed and traffic is highest for RClass5 and lowest for RClass1, a large variation in traffic and speed can be seen with a road class. These variations bring the spatial heterogeneity in emissions (Fig. 3 of the manuscript) due the heterogeneity in traffic and speed across roads of Delhi which reflects in the spatial emission analysis (Section 3.3). For eg. daytime average speed across all roads in Inner Delhi is 29 km/h which is lower than the daytime average speed of 32 km/h in outer Delhi. Moreover the emission flux over inner Delhi is higher as compared to outer Delhi indicating spatial variations driven by traffic and speed variation.

[Figure]

Figure 2. Boxplot for the PCU and speed variation across all road classes.

**Reference**

Hakkim, H., Kumar, A., Sinha, B., and Sinha, V.: Air pollution scenario analyses of fleet replacement strategies to accomplish reductions in criteria air pollutants and 74 VOCs over India, Atmospheric Environment: X, 13, 100150, https://doi.org/10.1016/j.aeaoa.2022.100150, 2022.

Sahu, S. K., Beig, G., and Parkhi, N. S.: Emissions inventory of anthropogenic PM2.5 and PM10 in Delhi during Commonwealth Games 2010, Atmospheric Environment, 45, 6180–6190, https://doi.org/10.1016/j.atmosenv.2011.08.014, 2011.

Malik, L., Tiwari, G., and Khanuja, R. K.: Classified Traffic Volume and Speed Study Delhi, Transportation Research and Injury Prevention Programme (TRIPP), http://tripp.iitd.ac.in/assets/publication/classified_volume_speed_studyDelhi-2018.pdf, 2018.

Malik, L., Tiwari, G., Biswas, U., and Woxenius, J.: Estimating urban freight flow using limited data: The case of Delhi, India, Transportation Research Part E: Logistics and Transportation Review, 149, 102316, https://doi.org/10.1016/j.tre.2021.102316, 2021.

Dhyani, R. and Sharma, N.: Sensitivity Analysis of CALINE4 Model under Mix Traffic Conditions, Aerosol Air Qual. Res., 17, 314–329, https://doi.org/10.4209/aaqr.2016.01.0012, 2017.